# DIFFUSION GENERATIVE FLOW SAMPLERS: IMPROVING LEARNING SIGNALS THROUGH PARTIAL TRAJECTORY OPTIMIZATION

**Dinghuai Zhang**[1,2][*] **Ricky T. Q. Chen**[3], **Cheng-Hao Liu**[1,4],
**Aaron Courville**[1,2] **& Yoshua Bengio**[1,2]
[1]Mila - Quebec AI Institute, [2]Université de Montréal, [3]FAIR, Meta AI, [4]McGill University

## ABSTRACT

We tackle the problem of sampling from intractable high-dimensional density functions, a fundamental task that often appears in machine learning and statistics. We extend recent sampling-based approaches that leverage controlled stochastic processes to model approximate samples from these target densities. The main drawback of these approaches is that the training objective requires full trajectories to compute, resulting in sluggish credit assignment issues due to use of entire trajectories and a learning signal present only at the terminal time. In this work, we present **D**iffusion **G**enerative **F**low **S**amplers (DGFS), a sampling-based framework where the learning process can be tractably broken down into short partial trajectory segments, via parameterizing an additional "flow function". Our method takes inspiration from the theory developed for generative flow networks (GFlowNets), allowing us to make use of intermediate learning signals. Through various challenging experiments, we demonstrate that DGFS achieves more accurate estimates of the normalization constant than closely-related prior methods.

## 1 INTRODUCTION

While diffusion models for generative modeling can be trained with efficient simulation-free training algorithms, their application in solving sampling problems—where no data is given but only an unnormalized density function—remains challenging. This is a very general problem formulation that appears across a diverse range of research fields including machine learning (ML) (Duane et al., 1987; Neal, 1995; Hernández-Lobato & Adams, 2015) and statistics (Neal, 1998; Andrieu et al., 2004; Zhang et al., 2021), and has wide application in scientific fields such as physics (Wu et al., 2018; Albergo et al., 2019; Noé et al., 2019) and chemistry (Frenkel & Smit, 1996; Hollingsworth & Dror, 2018; Holdijk et al., 2022).

Two main lines of work to tackle this sampling problems are Monte Carlo (MC) methods and variational inference (VI). MC methods achieve sampling typically via a carefully designed Markov chain (MCMC) or ancestrally sample from sequential targets such as a series of annealed distributions (Moral & Doucet, 2002). Despite their wide application, MC methods can easily get stuck in local modes or suffer from long mixing time (Tieleman & Hinton, 2009; Desjardins et al., 2010). On the other hand, VI methods (Blei et al., 2016) use a parametric model to approximate the target distribution by minimizing some divergence. However, these models often possess limited expressiveness, and optimizing the divergence can lead to bad learning dynamics (Minka, 2001).

Recent works have formulated sampling from unnormalized density functions as stochastic optimal control problems, where a diffusion model is trained as a stochastic process to generate the target distribution. At present, the difficulty in using these algorithms is in part due to the need to sequentially sample long trajectories during training, while only the final step is actually treated as a sample from the model. The popular Kullback–Leibler (KL) problem formulation only provides a learning signal for the terminal state, which can cause credit assignment problems in learning, especially for diffusion models which require fairly long sequential chains for sampling.

---

[*]Correspondence to `dinghuai.zhang@mila.quebec`

In this work, we build on top of Lahlou et al. (2023), who noted that recent methods that train diffusion models given an unnormalized density function (Zhang & Chen, 2022; Vargas et al., 2023a) can be interpreted from a GFlowNet perspective (Bengio et al., 2023). Although most GFlowNet works perform amortized inference on combinatorial discrete domains, previous work (Lahlou et al., 2023) establishes a theoretical foundation for continuous space GFlowNets and conducts numerical analysis to show their feasibility. Based on this framework, we propose the Diffusion Generative Flow Sampler (DGFS), an enhanced training method of diffusion models for sampling when given an unnormalized density function. In particular, we formulate a training approach that learns intermediate "flow[1] functions" that aggregate information for training intermediate states. This allows us to further define partial trajectory-based training objectives, where we have access to only partial specification of the sampling path. We find that this approach significantly reduces gradient noise and helps stabilize convergence. Furthermore, our training framework enables us to receive learning signals at intermediate steps, even before the sequential sampling is completed, in contrast to previous methods that can only receive signals at the terminal steps.

In summary, our contributions are as follows,

- We propose DGFS, an effective algorithm that trains stochastic processes to sample from given unnormalized target densities.
- We show that, contrary to other diffusion-based sampling algorithms, DGFS can (1) update its parameters without having full trajectory specification; (2) receive intermediate signals even before a sampling path is finished.
- We show through extensive empirical study that these special advantages enable DGFS to benefit from more stable and informative training signals, with DGFS generating samples accurately from the target distribution.

## 2 PRELIMINARIES

### 2.1 SAMPLING AS STOCHASTIC OPTIMAL CONTROL

We aim to sample from a $D$-dimensional target distribution $\pi(\cdot)$ in $\mathbb{R}^D$ defined by a given unnormalized target density $\mu(\cdot)$ where $\pi(\cdot) = \frac{\mu(\cdot)}{Z}$ and $Z = \int \mu(\mathbf{x}) \, d\mathbf{x}$. Following a similar setup to prior works such as Tzen & Raginsky (2019); Zhang & Chen (2022); Vargas et al. (2023a), we consider a sequential latent variable model with a joint distribution denoted $\mathcal{Q}$ of which samples follow a Markov process,

$$\mathcal{Q}(\mathbf{x}_0, \ldots, \mathbf{x}_N): \quad \mathbf{x}_{n+1} \sim P_F(\cdot|\mathbf{x}_n), \quad \mathbf{x}_0 \sim p_0^{\text{ref}}(\mathbf{x}_0). \tag{1}$$

Through this sequence model, we use the marginal distribution $\mathcal{Q}(\mathbf{x}_N)$ at the terminal step $N$ to approximate $\pi(\mathbf{x}_N)$. Here $p_0^{\text{ref}}(\cdot)$ is a reference distribution at time $n = 0$, and $P_F(\cdot|\cdot)$ denotes a forward transition probability (or policy) distribution. This forward policy depends on $n$, *i.e.* it is $P_F^{n+1|n}(\cdot|\mathbf{x}_n)$, but we omit the step index $n$ out of simplicity or assume $n$ is encoded in $\mathbf{x}_n$. We thus refer to $\mathcal{Q}$ as the *forward process* in the following text[2]. In this work, we constrain the transition to be a unimodal Gaussian distribution:

$$P_F(\cdot|\mathbf{x}_n) = \mathcal{N}(\cdot; \mathbf{x}_n + h\mathbf{f}(\mathbf{x}_n, n), h\sigma^2\mathbf{I}), \tag{2}$$

where $\mathbf{f}$ is a control drift parameterized by a neural network and $h$ is a constant step size. We also define a *reference process* with the forward policy $P_F^{\text{ref}}(\cdot|\mathbf{x}_n) = \mathcal{N}(\cdot; \mathbf{x}_n, h\sigma^2\mathbf{I})$,

$$\mathcal{Q}^{\text{ref}}(\mathbf{x}_0, \ldots, \mathbf{x}_N): \quad \mathbf{x}_{n+1} \sim P_F^{\text{ref}}(\cdot|\mathbf{x}_n), \quad \mathbf{x}_0 \sim p_0^{\text{ref}}(\mathbf{x}_0). \tag{3}$$

Since $P_F^{\text{ref}}$ has no nonlinear $\mathbf{f}$, the marginal distributions of $\mathcal{Q}^{\text{ref}}$ at step $n$ are known in closed form, denoted $p_n^{\text{ref}}(\cdot)$. We can then construct a *target process* that corresponds to this reference process,

$$\mathcal{P}(\mathbf{x}_0, \ldots, \mathbf{x}_N) := \mathcal{Q}^{\text{ref}}(\mathbf{x}_0, \ldots, \mathbf{x}_N) \frac{\pi(\mathbf{x}_N)}{p_N^{\text{ref}}(\mathbf{x}_N)}. \tag{4}$$

---

[1] In this work, the term "flow" specifically denotes the flow network structure in GFlowNets, which has nothing to do with normalizing flow models.

[2] Note that this is consistent with prior GFlowNet literature, though it is the opposite direction considered by standard diffusion model literature in generative modeling, where "forward process" is colloquially used to denote the process from clean sample to white noise.

It can be shown that the marginal distribution of $\mathcal{P}$ at the terminal time $N$ is exactly proportional to the target $\mu(\cdot)$ by marginalizing out the variables $\mathbf{x}_0, \ldots, \mathbf{x}_{N-1}$ in $\mathcal{P}$; see Theorem 1 of Zhang & Chen (2022). Hence it is natural to consider learning the drift $\mathbf{f}(\cdot, \cdot)$ in Equation 2 by minimizing the KL divergence $\mathcal{D}_{\mathrm{KL}}(\mathcal{Q}\|\mathcal{P})$ between the forward process and this target process, which is equivalent to the following optimal control problem:

$$\mathcal{Q}(\mathbf{x}_{0:N}): \quad \mathbf{x}_{n+1} = \mathbf{x}_n + h\mathbf{f}(\mathbf{x}_n, n) + \sqrt{h}\sigma\boldsymbol{\epsilon}_n, \quad \mathbf{x}_0 \sim p_0^{\mathrm{ref}}(\cdot), \tag{5}$$

$$\min_{\mathbf{f}} \mathbb{E}_{\mathcal{Q}} \left[ \sum_{n=0}^{N-1} \frac{h}{2\sigma^2} \|\mathbf{f}(\mathbf{x}_n, n)\|^2 + \log \Psi(\mathbf{x}_N) \right], \Psi(\cdot) := \frac{p_N^{\mathrm{ref}}(\cdot)}{\mu(\cdot)}, \tag{6}$$

where $\boldsymbol{\epsilon}_n \sim \mathcal{N}(\mathbf{0}, \mathbf{I}), \forall n$. See Appendix B.1 for derivation.

If we keep the total time $T = hN$ fixed and push the number of steps $N \to \infty$, this formulation recovers the continuous-time stochastic optimal control formulation seen in prior works (Zhang & Chen, 2022):

$$\mathcal{Q}(\mathbf{x}_{[0,T]}): \quad \mathrm{d}\mathbf{x}_t = \mathbf{f}(\mathbf{x}_t, t)\,\mathrm{d}t + \sigma\,\mathrm{d}\mathbf{W}_t, \quad \mathbf{x}_0 \sim p_0^{\mathrm{ref}}(\mathbf{x}_0), \tag{7}$$

$$\min_{\mathbf{f}} \mathbb{E}_{\mathcal{Q}} \left[ \int_0^T \frac{1}{2\sigma^2} \|\mathbf{f}(\mathbf{x}_t, t)\|^2\,\mathrm{d}t + \log \Psi(\mathbf{x}_T) \right], \tag{8}$$

where the forward process is defined by a variance exploding stochastic differential equation (VE-SDE)[3], and $\mathrm{d}\mathbf{W}_t$ is a standard Wiener process. This formulation appears in standard stochastic optimal control theory (Kappen, 2005). The seminal path integral sampler (PIS) (Zhang & Chen, 2022) proposes to solve such optimal control problems with deep learning to tackle the sampling problem. Its follow-up, the denoising diffusion sampler (DDS) (Vargas et al., 2023a), takes a similar approach but adopts a variance preserving (VP) SDE rather than the variance exploding ones in Equation 5 and 7. We defer the derivation of VP modeling to Appendix B.2. The practical implementations of these prior works are performed in discrete time with constant step sizes, equivalent to the discrete formulation outlined above. Hence we also adopt the discrete-time formulation in the following sections for simplicity and better amenability to our proposed algorithm.

## 2.2 GFLOWNETS

Generative flow networks (Bengio et al., 2023, GFlowNets) are a family of amortized inference algorithms that treat the problem of generation with probability proportional to given rewards in an Markov decision process (MDP) as a sequential decision-making process.

Let $\mathcal{G} = (\mathcal{S}, \mathcal{A})$ be a directed acyclic graph (DAG), where the set of vertices $\mathbf{s} \in \mathcal{S}$ are referred to as states, and the set of directed edges $(\mathbf{s} \to \mathbf{s}') \in \mathcal{A} \subseteq \mathcal{S} \times \mathcal{S}$ represent the transitions between the states and are called actions. Given a designated initial state $\mathbf{s}_0$ with no incoming edges and a set of *terminal states* $s_N$ without outgoing edges, a *complete trajectory* can be defined as a sequence of states $\tau = (\mathbf{s}_0 \to \mathbf{s}_1 \to \ldots \to \mathbf{s}_N) \in \mathcal{T}$. We define the *forward policy* $P_F(\mathbf{s}'|\mathbf{s})$ as a distribution over the children of every non-terminal state $\mathbf{s}$, which induces a distribution over complete trajectories, $P_F(\tau) = \prod_{n=0}^{N-1} P_F(\mathbf{s}_{n+1}|\mathbf{s}_n)$. This in turn defines a marginal distribution over terminating states $\mathbf{x}$ in the sense that $P_T(\mathbf{x}) = \sum_{\tau \to \mathbf{x}} P_F(\tau)$, where we call $P_T(\cdot)$ the GFlowNet's *terminating distribution*. The aim of GFlowNet learning is to obtain a policy such that $P_T(\cdot) \propto R(\cdot)$, where $R(\cdot): \mathcal{X} \to \mathbb{R}_+$ is a given *reward function* or unnormalized density, where we do not know the normalizing factor $Z = \sum_{\mathbf{x}} R(\mathbf{x})$. We also use the trajectory *flow* function $F(\tau) = ZP_F(\tau)$ to incorporate the effect of the partition function. Accordingly, the *state flow* function is the marginalization of the trajectory flow function $F(\mathbf{s}) = \sum_{\tau \ni \mathbf{s}} F(\tau): \mathcal{S} \to \mathbb{R}_+$. Most previous GFlowNet works are about composite discrete objects such as molecules; on the other hand, Lahlou et al. (2023) provide theoretical support and initial numerical demonstration for GFlowNets on continuous and hybrid (continuous and discrete) spaces. Based on this, our work aims to further tackle more complex sampling problems in continuous domains, *i.e.* $\mathcal{S} \subseteq \mathbb{R}^D$.

---

[3]To avoid confusion, we would like to remark again that in literature the term "forward process" is usually used to describe the noising process, which is actually variance shrinking in PIS; see Equation 21.

**Detailed balance (DB).** DB is one of the most important GFlowNet training losses. Apart from the forward policy and the state flow function, DB also requires the practitioners to parameterize an additional *backward policy* $P_B(\mathbf{s}|\mathbf{s}';\boldsymbol{\theta})$, which is a distribution over the parents of any noninitial state $\mathbf{s}' \in \mathcal{S}$. The DB objective is defined for a single transition $(\mathbf{s} \rightarrow \mathbf{s}')$ as follows,

$$\ell_{\text{DB}}(\mathbf{s}, \mathbf{s}'; \boldsymbol{\theta}) = \left( \log \frac{F(\mathbf{s}; \boldsymbol{\theta}) P_F(\mathbf{s}'|\mathbf{s}; \boldsymbol{\theta})}{F(\mathbf{s}'; \boldsymbol{\theta}) P_B(\mathbf{s}|\mathbf{s}'; \boldsymbol{\theta})} \right)^2. \tag{9}$$

According to the theory of GFlowNets (Bengio et al., 2023), if the DB loss reaches 0 for any transition pair, then the forward policy samples correctly from the target distribution defined by $R(\cdot)$.

## 3 DIFFUSION GENERATIVE FLOW SAMPLERS

### 3.1 AMORTIZING TARGET INFORMATION INTO INTERMEDIATE STEPS

Like many other probabilistic methods, the training objective of PIS is a KL divergence between the model and the target distribution (Equation 5). In particular, PIS's KL is calculated at the *trajectory* level, which means each computation relies on a complete trajectory $\tau = (\mathbf{x}_0, \ldots, \mathbf{x}_N)$ sampled from the current model's forward process. Since the training signal $\mu(\mathbf{x}_N)$ is only affected by the terminal state $\mathbf{x}_N$, long trajectories can potentially yield less informative credit assignment for the intermediate steps, which manifests as high gradient variance and can sometimes lead to more difficult optimization. Therefore, we propose to exploit *intermediate learning signals that do not necessarily rely on the complete trajectory information*. What if we could know or approximate the target distribution at any step $n$? Note that the target process (Equation 4) can be decomposed into a product of conditional distributions

$$\mathcal{P}(\mathbf{x}_{0:N}) = \pi(\mathbf{x}_N) \prod_{n=0}^{N-1} P_B(\mathbf{x}_n|\mathbf{x}_{n+1}), \tag{10}$$

where $P_B(\cdot|\cdot)$ denotes the backward transition probability (or backward policy) derived from the joint of the target process. Then we have the expression for the marginal distribution at step $n$:

$$p_n(\mathbf{x}_n) = \int \pi(\mathbf{x}_N) \prod_{l=n}^{N-1} P_B(\mathbf{x}_l|\mathbf{x}_{l+1}) \, \mathrm{d}\mathbf{x}_{n+1:N}, \forall n \in \{0, \ldots, N\}, \tag{11}$$

where we set $p_N(\cdot) = \pi(\cdot)$ for consistency. If we know the form of $p_n(\cdot)$ then we could directly learn from it with shorter trajectories and thus achieve more efficient training. Unfortunately, although we know the form of $P_B(\cdot|\cdot)$, for general target distribution there is generally no known analytical expression for it. As a result, we propose to use a deep neural network $F_n(\cdot; \boldsymbol{\theta})$ with parameter $\boldsymbol{\theta}$ as a "helper" to approximate the unnormalized density of the $n$-th step target $p_n(\cdot)$.

One naive way to train the $F_n(\cdot; \boldsymbol{\theta})$ network is to fit its value to the MC quadrature estimate of the integral in Equation 11. However, every training step would then have to include such computationally expensive quadrature calculation, making the resulting algorithm extremely inefficient. Instead, we propose turning this quadrature calculation into an optimization problem in an amortized way, and instead train $F_n(\cdot; \boldsymbol{\theta})$ to achieve the following constraint:

$$F_n(\mathbf{x}_n; \boldsymbol{\theta}) \prod_{l=n}^{N-1} P_F(\mathbf{x}_{l+1}|\mathbf{x}_l; \boldsymbol{\theta}) = \mu(\mathbf{x}_N) \prod_{l=n}^{N-1} P_B(\mathbf{x}_l|\mathbf{x}_{l+1}), \tag{12}$$

for all partial trajectories $\mathbf{x}_{n:N}$. Here, we would parameterize both the function $F_n$ and the distribution $P_F$ from Equation 2 by deep neural networks. Once Equation 12 is achieved, integrating $\mathbf{x}_{n+1:N}$ on both sides would lead to the fact that $F_n(\mathbf{x}_n; \boldsymbol{\theta})$ equals the integral in Equation 11. To put it another way, $F_n(\cdot; \boldsymbol{\theta})$ amortizes the integration computation into the learning of $\boldsymbol{\theta}$. Notice that we only use $\mu(\cdot)$ in Equation 12 because our problem setting provides only $\mu$. The unknown normalization constant is thus also absorbed into $F_n$. We construct a training objective by simply regressing the left hand side of Equation 12 to the right hand side. Furthermore, to ensure training stability, this regression is performed in the log space. In practice, the flow function at different steps share the same set of parameters; this is achieved by introducing an additional step embedding input for the $F(\cdot, n; \boldsymbol{\theta})$ neural network. As a result, our method learns two neural networks (the forward policy, which is the same as in PIS, and the flow function) in the meantime.

### 3.2 UPDATING PARAMETERS WITH INCOMPLETE TRAJECTORIES

If we compare Equation 12 with $n$ and $n + 1$, we obtain a formula with no dependence on $\mu$:

$$F_n(\mathbf{x}_n; \boldsymbol{\theta})P_F(\mathbf{x}_{n+1}|\mathbf{x}_n; \boldsymbol{\theta}) = F_{n+1}(\mathbf{x}_{n+1}; \boldsymbol{\theta})P_B(\mathbf{x}_n|\mathbf{x}_{n+1}), \tag{13}$$

where the details are deferred to Appendix C.1.

**GFlowNet perspective.** This formulation is similar to the GFlowNet DB loss in Equation 9. Actually, it is not hard to see that the above modeling is a variant of GFlowNets. As Zhang et al. (2022a) has pointed out and as later adopted by Lahlou et al. (2023), there is a connection between diffusion modeling and GFlowNets. In the GFlowNet language, the target density $\mu(\cdot)$ is the terminal reward function, the temporal-spatial specification space $\{(n, \mathbf{x}_n) : n \in \mathbb{N}, \mathbf{x}_n \in \mathbb{R}^D\}$ is the GFlowNet state space $\mathcal{S}$, the forward transition probability $P_F(\cdot|\cdot)$ of the diffusion process $\mathcal{Q}$ is the GFlowNet forward policy, the backward transition probability $P_B(\cdot|\cdot)$ of the reference process $\mathcal{Q}^{\text{ref}}$ is the GFlowNet backward policy, and the helper network $F_n(\cdot)$ is the state flow $F(\mathbf{s})$. Notice here that the backward policy is fixed and does not contain learnable parameters. In our setting, we use the step index $n$ to distinguish between intermediate states $\{(n, \mathbf{x}_n) : n < N, \mathbf{x}_n \in \mathbb{R}^D\}$ and terminating states $\{(N, \mathbf{x}_N), \mathbf{x}_N \in \mathbb{R}^D\}$[4]. What's more, the goal of GFlowNets, which is to generate terminal states with probability proportional to the reward value, matches the goal of sampling in our setting.

It is thus worthwhile to investigate how GFlowNets could be trained in this context. Apart from the DB constraint, Madan et al. (2022) propose a more general subtrajectory balance constraint: $F_m(\mathbf{x}_m; \boldsymbol{\theta}) \prod_{l=m}^{n-1} P_F(\mathbf{x}_{l+1}|\mathbf{x}_l; \boldsymbol{\theta}) = F_n(\mathbf{x}_n; \boldsymbol{\theta}) \prod_{l=m}^{n-1} P_B(\mathbf{x}_l|\mathbf{x}_{l+1})$, where $\mathbf{x}_{m:n}$ is a subtrajectory. Notice that Equation 12 is actually a special case of this constraint if we specify $n = N$. The resulting objective for our method is

$$\ell_{\text{SubTB}}(\mathbf{x}_{m:n}; \boldsymbol{\theta}) = \left( \log \frac{F_m(\mathbf{x}_m; \boldsymbol{\theta}) \prod_{l=m}^{n-1} P_F(\mathbf{x}_{l+1}|\mathbf{x}_l; \boldsymbol{\theta})}{F_n(\mathbf{x}_n; \boldsymbol{\theta}) \prod_{l=m}^{n-1} P_B(\mathbf{x}_l|\mathbf{x}_{l+1})} \right)^2. \tag{14}$$

A training loss based on transitions and more generally partial trajectories makes it possible to *update parameters without entire trajectory specification*. This can significantly reduce the variance in the optimization, as shown by Madan et al. (2022) in a similar context. To better combine the training effect from subtrajectories with different lengths, the adopted objective is

$$\mathcal{L}(\tau; \boldsymbol{\theta}) = \frac{\sum_{0 \le m < n \le N} \lambda^{n-m} \ell_{\text{SubTB}}(\mathbf{x}_{m:n})}{\sum_{0 \le m < n \le N} \lambda^{n-m}}, \tau = (\mathbf{x}_0, \dots, \mathbf{x}_N), \tag{15}$$

where $\lambda \in \mathbb{R}_+$ is a scalar for controlling assigned weights to different partial trajectories. For the modeling of the drift of the forward stochastic process, we use $\mathbf{f}(\cdot, n)/\sigma = \text{NN}_1(\cdot, n), +\text{NN}_2(n) \cdot \nabla \log \mu(\cdot)$, which includes two different neural networks and gradient information from unnormalized target densities in a similar form to Langevin dynamics (Zhang & Chen, 2022). We have conduct extensive ablation study about the parameterization, training objectives, exploration techniques, and other considerations, and display them in Appendix C.

**Improved credit assignment with local signals.** In both PIS and DDS algorithms, the only learning signal $\mu(\mathbf{x}_N)$ comes at the end of complete trajectories $\mathbf{x}_{0:N}$ when the density function is queried with $\mathbf{x}_N$. This is also the case for general GFlowNet methods, where we set $F_N(\mathbf{x}) \leftarrow \mu(\mathbf{x}), \forall \mathbf{x} \in \mathbb{R}^D$. In this way, the propagation of density information from terminal states to early states would be slow, thus affecting the training efficiency. We thus adopt the forward-looking trick proposed by Pan et al. (2023a) to incorporate intermediate learning signals for GFlowNets. The forward-looking technique essentially parameterizes the logarithm of the state flow function as $\log F_n(\mathbf{x}_n; \boldsymbol{\theta}) = \log \tilde{R}_n(\mathbf{x}_n) + \text{NN}(\mathbf{x}_n; \boldsymbol{\theta})$, where $\tilde{R}_n(\mathbf{x}_n)$ could be an arbitrary "forward-looking" estimate of how much energy $\mathbf{x}_n$ could contribute to the final reward $\mu(\mathbf{x}_N)$. The details of the "NN" neural network are deferred to Appendix C.4. In the proposed DGFS algorithm, we take the form of the (log) partial reward to be

$$\log \tilde{R}_n(\cdot) = (1 - \tfrac{n}{N}) \log p_n^{\text{ref}}(\cdot) + \tfrac{n}{N} \log \mu(\cdot), \tag{16}$$

which combines information from the target density and the reference marginal density.

---

[4]Without loss of generality, we slightly overuse the notations and simply use $\mathbf{x}_n$ to denote $(n, \mathbf{x}_n)$.

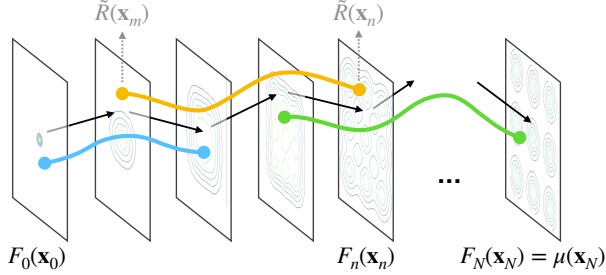
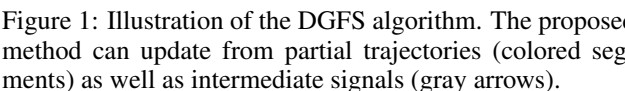
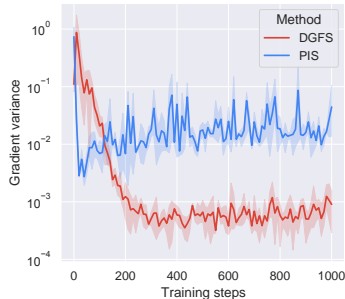

Figure 1: Illustration of the DGFS algorithm. The proposed method can update from partial trajectories (colored segments) as well as intermediate signals (gray arrows).

Figure 2: Gradient variance of DGFS and PIS, explaining the better training effects of DGFS.

We illustrate the resulting DGFS algorithm in Figure 1. Different from prior works, DGFS can update parameters with incomplete subtrajectories (colored curve segments in the figure) with the help of amortized flow functions, as well as intermediate signals at non-terminal step (grey arrows in the figure), which fundamentally improves the credit assignment process.

**Estimation of the partition function.** For an arbitrary trajectory $\tau = \{\mathbf{x}_n\}_{n=0}^N$, one could define its (log) importance weight to be $S(\tau; \boldsymbol{\theta}) = \log \mathcal{P}(\mathbf{x}_{0:N}) - \log \mathcal{Q}(\mathbf{x}_{0:N}; \boldsymbol{\theta})$, where $\mathcal{Q}(\mathbf{x}_{0:N}; \boldsymbol{\theta}) = p_0^{\text{ref}}(\mathbf{x}_0) \prod_{n=0}^{N-1} P_F(\mathbf{x}_{n+1}|\mathbf{x}_n; \boldsymbol{\theta})$. We obtain the following log-partition function ($\log Z$) estimator:

$$\log \sum_{b=1}^{B} \exp\left(S(\tau^{(b)}; \boldsymbol{\theta})\right) - \log B \leq \log Z, \quad \tau^{(b)} \sim \mathcal{Q}(\cdot; \boldsymbol{\theta}), \tag{17}$$

where $B \in \mathbb{N}_+$ is the number of particles (*i.e.*, trajectories) used for the estimation. Similar to the importance weighted auto-encoders (Burda et al., 2016), our estimator provides a valid lower bound of the logarithm of the normalizing factor with *any* forward process $\mathcal{Q}$. In this work, we use the difference between estimated log partition function and the ground truth as an evaluation metric. See the Appendix C.2 for additional analysis. We also remark that this approach is similar to the neural importance sampling in Nicoli et al. (2020); Wirnsberger et al. (2020) in the sense that we use the forward process $\mathcal{Q}$ as the proposal distribution.

## 3.3 DISCUSSION

**Variance of gradient updates.** Figure 2 shows that the variance of stochastic gradients with DGFS is much smaller than that with PIS; notice that the comparison is valid as we take the same neural network architecture for both algorithms. We hypothesize that DGFS benefits from the low variance behavior due to the following two reasons: (1) From the perspective of temporal difference learning (Sutton & Barto, 2018; Kearns & Singh, 2000), it has long been discussed that policy learning from shorter trajectory data would lower the variance at the cost of potentially higher bias. Before this work, it was impossible for sampling with stochastic process algorithms to update with partial trajectories, while in the proposed DGFS method, we use an additional flow network to make this feasible. In Figure 3 we also show that the flow network can be properly learned and thus does not introduce much bias, indicating that DGFS achieves a better bias-variance trade-off. (2) Probabilistic ML researchers have shown that for KL divergence based distribution learning, the gradient will not vanish even if the optimal distribution has been reached; see Roeder et al. (2017) and Section 4 of Xu et al. (2022). Later works including Vaitl et al. (2022a;b) propose a different path-gradient estimator to tackle this issue with a similar insight with Roeder et al. (2017); see the related work section. On the other hand, the DGFS objective does achieve zero gradient with the optimal solution (*c.f.* Appendix C.3), making it a more stable algorithm.

**Modeling considerations.** Notice that the DGFS derivation in Section 3 does not rely on the specific form of the stochastic process. Therefore, DGFS could take both the variance exploding (VE) (Zhang & Chen, 2022) or variance preserving (VP) (Vargas et al., 2023a) formulation. In our numerical study, we find the former to be more stable, and thus we default to it unless otherwise specified. We defer related ablations to the Appendix C.5.

Table 1: Experimental results on benchmarking target densities. For each algorithm, we report its absolute estimation bias for the log normalizing factor and the corresponding standard deviation. Our method achieves the best performance among the diffusion modeling-based samplers.

| | MoG | Funnel | Manywell | VAE | Cox |
|---|---|---|---|---|---|
| SMC | $0.289_{\pm 0.112}$ | $0.307_{\pm 0.076}$ | $22.36_{\pm 7.536}$ | $14.34_{\pm 2.604}$ | $99.61_{\pm 8.382}$ |
| VI-NF | $1.354_{\pm 0.473}$ | $0.272_{\pm 0.048}$ | $2.677_{\pm 0.016}$ | $6.961_{\pm 2.501}$ | $83.49_{\pm 2.434}$ |
| CRAFT | $0.348_{\pm 0.210}$ | $0.454_{\pm 0.485}$ | $0.987_{\pm 0.599}$ | $0.521_{\pm 0.239}$ | $13.79_{\pm 2.351}$ |
| FAB w/ Buffer[5] | $0.003_{\pm 0.0005}$ | $0.0022_{\pm 0.0005}$ | $0.032_{\pm 0.004}$ | N/A | $0.19_{\pm 0.04}$ |
| PIS | $0.036_{\pm 0.007}$ | $0.305_{\pm 0.013}$ | $1.391_{\pm 1.014}$ | $2.049_{\pm 2.826}$ | $11.28_{\pm 1.365}$ |
| DDS | $0.028_{\pm 0.013}$ | $0.416_{\pm 0.094}$ | $1.154_{\pm 0.626}$ | $1.740_{\pm 1.158}$ | N/A[6] |
| DGFS | $0.019_{\pm 0.008}$ | $0.274_{\pm 0.014}$ | $0.904_{\pm 0.067}$ | $0.180_{\pm 0.083}$ | $8.974_{\pm 1.169}$ |

**Convergence guarantees.** Previous theoretical studies (Bortoli, 2022; Chen et al., 2022; Lee et al., 2022) have demonstrated that the distribution of terminal samples generated by a diffusion model can converge to the target distribution under mild assumptions if the control term is well learned. These findings are also applicable to DGFS since these proofs are independent of how the networks are trained. Additionally, Zhang et al. (2022a) have proven that a perfectly learned score term corresponds to a zero GFlowNet training loss under mild conditions, following theory of GFlowNets (Bengio et al., 2021; 2023). This provides assurance that a sufficiently well-trained DGFS can accurately sample from the target distribution.

## 4 RELATED WORKS

**Sampling methods.** MCMC methods (Brooks et al., 2011) are the most classical sampling algorithms, including Langevin dynamics (Welling & Teh, 2011), Hamiltonian Monte Carlo (Duane et al., 1987; Hoffman & Gelman, 2011; Chen et al., 2014), and others (Huang et al., 2023). With MCMC operators as building blocks, sequential MC methods (Liu, 2001; Doucet et al., 2001) generate samples ancestrally through a series of annealed targets. On the other hand, people turn to parametric models such as deep networks to learn samplers via variational inference (Rezende & Mohamed, 2015). One the most commonly adopted models is the normalizing flow (Dinh et al., 2014) due to its tractability and expressiveness. Various work have been conducted to improve normalizing flow based sampling (Müller et al., 2018; Gao et al., 2020; Wu et al., 2020; Chen et al., 2020; Gabri'e et al., 2021; Arbel et al., 2021; Midgley et al., 2022; de G. Matthews et al., 2022; Caselle et al., 2022; Gao et al., 2023). These methods support asymptotically unbiased estimation as shown in Nicoli et al. (2020). What's more, Vaitl et al. (2022a) propose to to improve the gradient estimation of normalizing flow parameters with the path-gradient technique; Vaitl et al. (2022b) improve continuous normalizing flows based methods (Chen et al., 2018) with a similar technique. These methods have wide applications in natural sciences, for example in lattice field theories (Nicoli et al., 2021; 2023) and targeted free energy estimation (Wirnsberger et al., 2020).

Recently, many researchers have developed powerful diffusion process to model distributions in the generative modeling community (Vincent, 2011; Sohl-Dickstein et al., 2015; Ho et al., 2020a; Song et al., 2020; Du et al., 2022). This inspires people to study how to utilize diffusion process as samplers to learn distribution from given unnormalized target densities (Tzen & Raginsky, 2019; Zhang & Chen, 2022; Vargas et al., 2021; Geffner & Domke, 2022; Doucet et al., 2022; Vargas et al., 2023a; Richter et al., 2023; Vargas & Nusken, 2023; Vargas et al., 2023c;b). Our work also falls into this category. Specifically, Vargas et al. (2023a) can be seen as a special case of the method proposed in Berner et al. (2022).

**GFlowNets.** GFlowNet (Bengio et al., 2023) is a family of generalized variational inference algorithms with a focus on treating the sampling process as a sequential decision-making process.

---

[5]We used the fab-jax results. The FAB method utilizes the annealed importance sampling computation during the training period.

[6]Due to time constraints, we are unable to reproduce DDS performance using PyTorch for this task. Instead, we encountered "not a number", probably due to a mismatch in implementation details or other unspecified difference between PyTorch and Jax. The DDS results reported in its original paper can be found here.

It is original proposed to sample diversity-seeking objects in structured scientific domains such as biological seuqence and molecule design (Bengio et al., 2021; Jain et al., 2022; 2023b;a; Shen et al., 2023; Zhang et al., 2023c; Hernández-García et al., 2023; Kim et al., 2023). However, people have developed GFlowNet theories and methodologies to be suitable for generating general objects with composite structures. GFlowNet has rich connection with existing probabilistic ML methods (Zhang et al., 2022b; Zimmermann et al., 2022; Malkin et al., 2023; Zhang et al., 2022a; Hu et al., 2023; Ma et al.). Malkin et al. (2023) also show that GFlowNet has the capability of training with off-policy data compared to previous amortized inference methods. As a policy learning framework under Markov decision processes (MDPs), researchers have also drawn fruitful connections between GFlowNet and reinforcement learning (Pan et al., 2023b;c;a). GFlowNet has also been demonstrated to have wide application in causal discovery (Deleu et al., 2022; 2023; Atanackovic et al., 2023), feature attribution (Li et al., 2023a), network structure learning (Liu et al., 2022), job scheduling (Zhang et al., 2023a), and graph combinatorial optimization (Zhang et al., 2023b). Most prior works focus on structured discrete data space. Lahlou et al. (2023) first provides a sound theoretical framework and initial numerical demonstration for using GFlowNet on continuous space. Our work follows this framework and scales up the performance. A prior study (Li et al., 2023b) initially explored GFlowNets in a continuous space, whereas their theory is demonstrated to have non-trivial flaws when considering integration under continuous measure as indicated by Lahlou et al. (2023).

## 5 EXPERIMENTS

### 5.1 BENCHMARKING TARGET DISTRIBUTIONS

In this section, we conduct extensive experiments on a diverse set of challenging continuous sampling benchmarks to corroborate the effectiveness of DGFS against previous methods.

*Mixture of Gaussians (MoG)* is a 2-dimensional Gaussian mixture where there are 9 modes designed to be well-separated from each other. The modes share the same variance of $0.3$ and the means are located in the grid of $\{-5, 0, 5\} \times \{-5, 0, 5\}$.

*Funnel* is a classical sampling benchmark problem from Neal (2003); Hoffman & Gelman (2011). This 10-dimensional density is defined by $\mu(\mathbf{x}) = \mathcal{N}(x^{(0)}; 0, 9)\mathcal{N}(\mathbf{x}^{(1:9)}; \mathbf{0}, \exp(x^{(0)})\mathbf{I})$.[7]

*Manywell* (Midgley et al., 2022; Noé et al., 2019; Wu et al., 2020) is a 32-dimensional distribution defined by the product of 16 same 2-dimensional "double well" distributions, which share the unnormalized density of $\mu(x, y) = \exp(-x^4 + 6x^2 + 0.5x - 0.5y^2)$.

*VAE* is a 30-dimensional sampling task where we use the latent posterior of a pretrained variational autoencoder (Kingma & Welling, 2014; Rezende et al., 2014, VAE) to be the target distribution.

*Cox* denotes the 1600-dimensional target distribution of the log Gaussian cox process introduced in Møller et al. (1998) which models the positions of pine saplings. Its unnormalized density is $\mu(\mathbf{x}) = \mathcal{N}(\mathbf{x}; \mu, \mathbf{K}^{-1}) \prod_{d=1}^{D} \exp(x^{(d)}y^{(d)} - \alpha x^{(d)})$, where $\mu, \mathbf{K}, \mathbf{y}, \alpha$ are constants.

**Baselines.** We benchmark DGFS performance against a wide range of strong baseline methods. As an MCMC methods we include the sequential Monte Carlo sampler (Moral & Doucet, 2002, SMC), which is often considered the state-of-the-art sampling method. As for normalizing flow methods we include the variational inference with normalizing flows (Rezende & Mohamed, 2015, VI-NF) approach as well as more recent de G. Matthews et al. (2022, CRAFT) which is combined with MCMC. We also provide results of Midgley et al. (2022, FAB) with a replay buffer that utilizes the annealed importance sampling technique to augment the training. Last but not least, we compare with PIS (Zhang & Chen, 2022) and DDS (Vargas et al., 2023a) algorithms which are closest to our framework but trained with the KL divergence formulation (Equation 5) without the improved training techniques described in Section 3.

**Evaluation protocol.** We measure the performance of different algorithms by their estimation bias of the log normalizing factor of unnormalized target distributions. Contrary to Zhang & Chen (2022)

---

[7]Previous works (Zhang & Chen, 2022; Lahlou et al., 2023) unintentionally take the variance of $x^{(0)}$ to be 1 for their methods (hence much less difficulty) but use 9 for other methods.

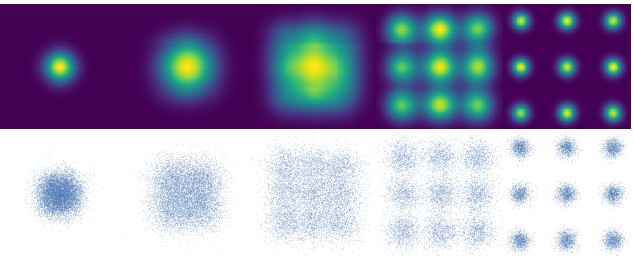

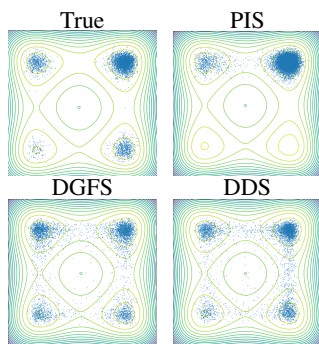

Figure 3: The learned DGFS flow function and the ground truth samples from target process at different diffusion steps. This shows DGFS is able to learn flow functions correctly.

Figure 4: Manywell plots. DGFS and DDS but not PIS recover all modes.

which take the best checkpoint across the run, in this work we compute the average of the last ten checkpoints for a more robust comparison. All results are conducted with 5 random seeds.

We display the benchmarking results in Table 1. We fail to achieve reasonable performance with DDS on Cox target where we mark it as "N/A". The advantage of the proposed DGFS method can be observed across different tasks, indicating it could indeed benefit from its design providing more informative training signals to achieve better convergence and capturing diverse modes even in high dimensional spaces. FAB with buffer produces highly stable results, which can be attributed to both annealed importance sampling, the use of buffers, larger networks, and other tricks. It is also remarkable that DGFS obtains very stable results compared to other baselines with diffusion modeling (PIS, DDS). We defer related experimental details and extra experiments to Appendix C.

## 5.2 ANALYSIS

First, we investigate the learning of the flow function introduced in Equation 12. In the first row of Figure 3, we visualize the learned flow function $F_n(\mathbf{x}_n)$ with $n = 20, 40, \ldots, 100$ in the 2D MoG task. In the second row, we simulate samples from $p_n(\mathbf{x}_n)$ by sampling according to Equation 11, *i.e.*, moving ground truth $\pi(\cdot)$ samples in the backward direction with policy $P_B$. The consistency between these two rows indicate that DGFS flow function can successfully approximate the intermediate marginal $p_n(\cdot)$, which makes it possible to provide an effective training signal for learning from partial trajectory pieces.

For the MoG task, we demonstrate the samples together with ground truth contours in Figure 5. This visualization result matches the log partition function estimation bias in Table 1, where DGFS generates the nine modes more uniformly than other baselines. This corroborates the advantage of DGFS to capture all different modes in a complex landscape. We then show visualization results on the Manywell task in Figure 4. The Manywell distribution is 32-dimensional, so we display the joint of the first and third dimension. We plot the samples generated from the true distribution and different algorithms, together with the true density contours. This figure implies that PIS can model half of the modes in these two dimensions well, but miss the other two modes, whereas both DGFS and DDS could cover all four modes. Further, DGFS achieves slightly better mode capturing than DDS in the sense that its modes are more separate and thus closer to the true distribution, while for DDS there are considerable samples appear between the upper right mode and the bottom right mode, which is not ideal. We defer other visualizations to the Appendix C.5.

To look into why DGFS outperforms PIS with similar architecture, we plot the learned drift network magnitude in Figure 6 in the Appendix. The result shows that the solution that DGFS achieved has more homogeneous magnitude outputs across different diffusion steps. This may explain one reason

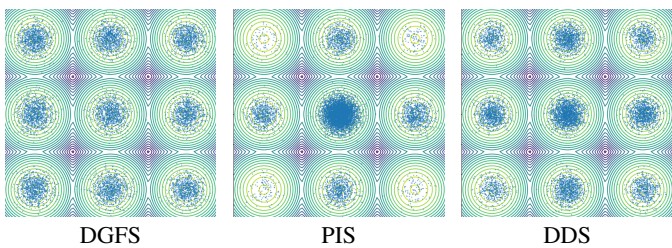

Figure 5: MoG visualization of DGFS and other diffusion-based samplers shows that DGFS could capture the diverse modes well. The contours display the landscape of the target density.

why DGFS is easier to train than
PIS, since the policy probabili-
ties at different steps $\{P_F(\cdot|\mathbf{x}_n)\}_n$ are actually parameterized by a single neural network with temporal embedding input. This makes the learning task more stationary with DGFS. We also conduct in-depth ablation studies to evaluate the design choices of DGFS in Appendix C.5 and C.7.

## 6 CONCLUSION

We propose the Diffusion Generative Flow Sampler (DGFS), a novel algorithm that trains diffusion models to sample from given unnormalized target densities. Different from prior works that could only learn from complete diffusion chains, DGFS could update its parameter with only partial specification of the stochastic process trajectory; what's more, DGFS can receive intermediate signals before completing the entire path. These features help DGFS benefit from efficient credit assignment and thus achieve better partition function estimation bias in the sampling benchmarks.

The DGFS framework presents various opportunities for further research, fueled by its inherent limitations. To name a few, how can we better design the intermediate local signals in ways more sophisticated than the current straightforward approach? Additionally, can we harness DGFS's exploration capabilities to succeed in high dimensional tasks, pushing the boundaries of its performance? Can we combine DGFS with a prioritized replay buffer? This would largely reduce the number of querying $\mu(\cdot)$. From the perspective of application, there are also many interesting research questions. Can we deploy DGFS on scientific biological or chemical tasks such as protein conformation modeling, with equivariant modeling? We expect that exploring these future directions will lead to valuable insights and bring fruit follow-ups of DGFS.

### ACKNOWLEDGEMENT

The authors are grateful to the discussions with Francisco Vargas, Julius Berner, Lorenz Richter, Laurence Midgley, Minsu Kim, Yihang Chen, and Qinsheng Zhang.

### ETHICS STATEMENT

We commit to the ICLR Code of Ethics and affirm that our work only utilizes public datasets for experimentation. While our empirical results are largely based on synthetic data, we acknowledge the potential for misuse and urge the responsible application of the proposed methods with real-world data. We welcome any related discussions and feedback.

### REPRODUCIBILITY STATEMENT

We provide detailed algorithmic and experimental description in Section 3 and Appendix C, and we have open sourced the code accompanying this research in this GitHub link.

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

## A  NOTATIONS

| Symbol | Description |
|---|---|
| $\mathbb{R}^D$ | $D$-dimensional real space |
| $N \in \mathbb{N}_+$ | number of steps in the discrete-time stochastic process |
| $\mathcal{S}$ | GFlowNet state space, equals $\{(n, \mathbf{x}_n) : n \in \{0, 1, \ldots, N\}, \mathbf{x}_n \in \mathbb{R}^D\}$; $(n, \mathbf{x}_n)$ is often writen as $\mathbf{x}_n$ for simplicity |
| $\mathcal{A}$ | action / transition space (edges $\mathbf{s} \to \mathbf{s}'$) |
| $\mathbf{s}$ | state in $\mathcal{S}$ |
| $\mathbf{s}_0$ | initial state, equals $(0, \mathbf{0})$ in VE modeling |
| $\tau$ | complete trajectory $(\mathbf{x}_0 \to \ldots \to \mathbf{x}_N)$, or $\mathbf{x}_{0:N}$ |
| $\boldsymbol{\theta}$ | learnable parameter of DGFS |
| $F_n : \mathbb{R}^D \to \mathbb{R}_+$ | state flow function at step $n$ |
| $P_F$ | forward policy (distribution over children / next step) |
| $P_B$ | backward policy (distribution over parents / previous step) |
| $\mathcal{Q}$ | forward process defined in Equation 1 |
| $\mathcal{Q}^{\text{ref}}$ | reference process defined in Equation 3 |
| $\mathcal{P}$ | target process defined in Equation 4 |
| $p_n^{\text{ref}} : \mathbb{R}^D \to \mathbb{R}_+$ | marginal distribution of reference process at step $n$ |
| $h \in \mathbb{R}_+$ | discretized step size for the stochastic process |
| $\mathbf{f}$ | drift term in the VE stochastic process (Equation 5) |
| $\sigma$ | variance coefficient in the stochastic process |
| $\mathrm{d}\mathbf{W}_t$ | standard Wiener process in continuous-time stochastic process |
| $R : \mathbb{R}^D \to \mathbb{R}_+$ | reward function (unnormalized target density) |
| $\tilde{R}_n : \mathbb{R}^D \to \mathbb{R}_+$ | intermediate forward-looking signal defined in Equation 16 |
| $Z \in \mathbb{R}_+$ | scalar, equal to $\sum_{\mathbf{x} \in \mathbb{R}^D} R(\mathbf{x})$ |
| $\mu : \mathbb{R}^D \to \mathbb{R}_+$ | same thing as $R(\cdot)$ |
| $\pi : \mathbb{R}^D \to \mathbb{R}_+$ | normalized density, equal to $\mu(\cdot)/Z$ |

## B  DETAILS ABOUT STOCHASTIC PROCESSES

### B.1  VARIANCE EXPLODING (VE) PROCESS

Song et al. (2020) and Zhang & Chen (2022) take a variance exploding stochastic process modeling. In Zhang & Chen (2022), the authors use Gisarnov theorem to derive an optimal control objective for KL divergence minimization. In this work, we present a discrete-time derivation for completeness.

The reference process starts at $p_0^{\text{ref}}(\cdot)$, which is a Dirac distribution at $\mathbf{0}$, and writes

$$\mathcal{Q}^{\text{ref}}(\mathbf{x}_{0:N}) : \ \mathbf{x}_{n+1} = \mathbf{x}_n + \sqrt{h}\sigma\boldsymbol{\epsilon}_n, \boldsymbol{\epsilon}_n \sim \mathcal{N}(\mathbf{0}, \mathbf{I}), \mathbf{x}_0 = \mathbf{0}, \tag{18}$$

$$\Rightarrow \mathcal{Q}^{\text{ref}}(\mathbf{x}_{0:N}) = p_0^{\text{ref}}(\mathbf{x}_0) \prod_{n=0}^{N-1} P_F^{\text{ref}}(\mathbf{x}_{n+1}|\mathbf{x}_n) = p_N^{\text{ref}}(\mathbf{x}_N) \prod_{n=0}^{N-1} P_B(\mathbf{x}_n|\mathbf{x}_{n+1}). \tag{19}$$

Therefore, $p_n^{\text{ref}}(\cdot) = \mathcal{N}(\cdot; \mathbf{0}, hn\sigma^2\mathbf{I})$. According to Bayesian formula,

$$P_B(\mathbf{x}_n|\mathbf{x}_{n+1}) = \frac{p_n^{\text{ref}}(\mathbf{x}_n)P_F^{\text{ref}}(\mathbf{x}_{n+1}|\mathbf{x}_n)}{p_{n+1}^{\text{ref}}(\mathbf{x}_{n+1})} = \frac{\mathcal{N}(\mathbf{x}_n; \mathbf{0}, hn\sigma^2\mathbf{I})\mathcal{N}(\mathbf{x}_{n+1}; \mathbf{x}_n, h\sigma^2\mathbf{I})}{\mathcal{N}(\mathbf{x}_{n+1}; \mathbf{0}, h(n+1)\sigma^2\mathbf{I})} \tag{20}$$

$$= \mathcal{N}(\mathbf{x}_n; \frac{n}{n+1}\mathbf{x}_{n+1}, \frac{n}{n+1}h\sigma^2\mathbf{I}). \tag{21}$$

The target process is then

$$\mathcal{P}(\mathbf{x}_0, \ldots, \mathbf{x}_N) = \mathcal{Q}^{\text{ref}}(\mathbf{x}_0, \ldots, \mathbf{x}_N) \frac{\pi(\mathbf{x}_N)}{p_N^{\text{ref}}(\mathbf{x}_N)} = \pi(\mathbf{x}_N) \prod_{n=0}^{N-1} P_B(\mathbf{x}_n|\mathbf{x}_{n+1}). \tag{22}$$

The trajectory level log probability difference between model and target is

$$\log \frac{\mathcal{Q}}{\mathcal{P}} = \log \frac{\mathcal{Q}}{\mathcal{Q}^{\text{ref}}} + \log \frac{\mathcal{Q}^{\text{ref}}}{\mathcal{P}}. \tag{23}$$

It is obvious that the second term is $\frac{\mathcal{Q}^{\text{ref}}}{\mathcal{P}} = \frac{p_N^{\text{ref}}(\mathbf{x}_N)}{\pi(\mathbf{x}_N)}$, thus we only need to look at the first term

$$\log \frac{\mathcal{Q}}{\mathcal{Q}^{\text{ref}}} = \sum_{n=0}^{N-1} \log \frac{P_F(\mathbf{x}_{n+1}|\mathbf{x}_n; \boldsymbol{\theta})}{P_F^{\text{ref}}(\mathbf{x}_{n+1}|\mathbf{x}_n)} = \sum_{n=0}^{N-1} \log \frac{\mathcal{N}(\cdot; \mathbf{x}_n + h\mathbf{f}(\mathbf{x}_n, n), h\sigma^2\mathbf{I})}{\mathcal{N}(\cdot; \mathbf{x}_n, h\sigma^2\mathbf{I})} \tag{24}$$

$$= \sum_{n=0}^{N-1} \left( \frac{h}{2\sigma^2} \|\mathbf{f}(\mathbf{x}_n, n; \boldsymbol{\theta})\|^2 + \frac{\sqrt{h}}{\sigma^2} \mathbf{f}(\mathbf{x}_n, n; \boldsymbol{\theta})^T \boldsymbol{\epsilon}_n \right). \tag{25}$$

Notice that under the expectation of $\mathcal{Q}$,

$$\mathbb{E}_{\mathcal{Q}} \left[ \mathbf{f}(\mathbf{x}_n, n; \boldsymbol{\theta})^T \boldsymbol{\epsilon}_n \right] = \mathbf{f}(\mathbf{x}_n, n; \boldsymbol{\theta})^T \mathbb{E}_{\mathcal{Q}} \left[ \boldsymbol{\epsilon}_n \right] = 0. \tag{26}$$

Accordingly,

$$\mathcal{D}_{\text{KL}}(\mathcal{Q}\|\mathcal{P}) = E_{\mathcal{Q}} \left[ \log \frac{\mathcal{Q}}{\mathcal{P}} \right] = E_{\mathcal{Q}} \left[ \sum_{n=0}^{N-1} \frac{h}{2\sigma^2} \|\mathbf{f}(\mathbf{x}_n, n; \boldsymbol{\theta})\|^2 + \log \frac{p_N^{\text{ref}}(\mathbf{x}_N)}{\pi(\mathbf{x}_N)} \right], \tag{27}$$

which leads us to Equation 6.

## B.2 VARIANCE PRESERVING (VP) PROCESS

Ho et al. (2020b) and Vargas et al. (2023a) take this VP formulation of diffusion process. The derivation is similar to the last section, just with a different variance schedule.

We first define the backward policy

$$P_B(\mathbf{x}_n|\mathbf{x}_{n+1}) = \mathcal{N}(\mathbf{x}_n; \sqrt{1 - \beta_n}\mathbf{x}_{n+1}; \beta_n\sigma^2\mathbf{I}). \tag{28}$$

Then The target and reference process are

$$\mathcal{P}(\mathbf{x}_{0:N}) = \pi(\mathbf{x}_N) \prod_{n=0}^{N-1} P_B(\mathbf{x}_n|\mathbf{x}_{n+1}), \tag{29}$$

$$\mathcal{Q}^{\text{ref}}(\mathbf{x}_{0:N}) = p_N^{\text{ref}}(\mathbf{x}_N) \prod_{n=0}^{N-1} P_B(\mathbf{x}_n|\mathbf{x}_{n+1}), \tag{30}$$

where $p_N^{\text{ref}}(\cdot) = \mathcal{N}(\cdot; \mathbf{0}, \sigma^2\mathbf{I})$. This guarantees that $p_n^{\text{ref}}(\cdot) = \mathcal{N}(\cdot; \mathbf{0}, \sigma^2\mathbf{I}), \forall n$.

From using Bayesian formula on $\mathcal{Q}^{\text{ref}}$,

$$P_F^{\text{ref}}(\mathbf{x}_{n+1}|\mathbf{x}_n) = \frac{p_{n+1}^{\text{ref}}(\mathbf{x}_{n+1}) P_B(\mathbf{x}_n|\mathbf{x}_{n+1})}{p_n^{\text{ref}}(\mathbf{x}_n)} = \mathcal{N}(\mathbf{x}_{n+1}; \sqrt{1 - \beta_n}\mathbf{x}_n, \beta_n\sigma^2\mathbf{I}). \tag{31}$$

Inspired by this, let us define the (learnable) forward process to be

$$P_F(\mathbf{x}_{n+1}|\mathbf{x}_n; \boldsymbol{\theta}) = \mathcal{N}(\mathbf{x}_{n+1}; \sqrt{1 - \beta_n}\mathbf{x}_n + \mathbf{f}(\mathbf{x}_n, n; \boldsymbol{\theta}), \beta_n\sigma^2\mathbf{I}), \tag{32}$$

$$\mathcal{Q}(\mathbf{x}_{0:N}) = p_0^{\text{ref}}(\mathbf{x}_0) \prod_{n=0}^{N-1} P_F(\mathbf{x}_{n+1}|\mathbf{x}_n; \boldsymbol{\theta}). \tag{33}$$

Similar to the derivation in VE,

$$\mathbb{E}_{\mathcal{Q}} \left[ \log \frac{\mathcal{Q}}{\mathcal{Q}^{\text{ref}}} \right] = \mathbb{E}_{\mathcal{Q}} \left[ \sum_{n=0}^{N-1} \log \frac{P_F(\mathbf{x}_{n+1}|\mathbf{x}_n; \boldsymbol{\theta})}{P_F^{\text{ref}}(\mathbf{x}_{n+1}|\mathbf{x}_n)} \right] = \sum_{n=0}^{N-1} E_{\mathcal{Q}} \left[ \frac{1}{2\beta_n\sigma^2} \|\mathbf{f}(\mathbf{x}_n, n; \boldsymbol{\theta})\|^2 \right]. \tag{34}$$

Therefore,

$$\mathcal{D}_{\text{KL}}(\mathcal{Q}\|\mathcal{P}) = E_{\mathcal{Q}} \left[ \log \frac{\mathcal{Q}}{\mathcal{P}} \right] = E_{\mathcal{Q}} \left[ \sum_{n=0}^{N-1} \frac{1}{2\beta_n\sigma^2} \|\mathbf{f}(\mathbf{x}_n, n; \boldsymbol{\theta})\|^2 + \log \frac{p_N^{\text{ref}}(\mathbf{x}_N)}{\pi(\mathbf{x}_N)} \right], \tag{35}$$

which is the learning objective of DDS algorithm.

## C ALGORITHMIC DETAILS

### C.1 RE-DERIVING DETAILED BALANCE CONSTRAINT

Writing down Equation 12 with $n$ and $n + 1$, we have

$$F_n(\mathbf{x}_n; \boldsymbol{\theta}) \prod_{l=n}^{N-1} P_F(\mathbf{x}_{l+1}|\mathbf{x}_l; \boldsymbol{\theta}) = \mu(\mathbf{x}_N) \prod_{l=n}^{N-1} P_B(\mathbf{x}_l|\mathbf{x}_{l+1}), \tag{36}$$

$$F_{n+1}(\mathbf{x}_{n+1}; \boldsymbol{\theta}) \prod_{l=n+1}^{N-1} P_F(\mathbf{x}_{l+1}|\mathbf{x}_l; \boldsymbol{\theta}) = \mu(\mathbf{x}_N) \prod_{l=n+1}^{N-1} P_B(\mathbf{x}_l|\mathbf{x}_{l+1}), \tag{37}$$

$$\Rightarrow \frac{F_n(\mathbf{x}_n; \boldsymbol{\theta}) P_F(\mathbf{x}_{n+1}|\mathbf{x}_n; \boldsymbol{\theta})}{F_{n+1}(\mathbf{x}_{n+1}; \boldsymbol{\theta})} = P_B(\mathbf{x}_n|\mathbf{x}_{n+1}) \tag{38}$$

where the last equation is obtained by comparing the first two equations on both sides, which resonates the GFlowNet DB criterion.

### C.2 $\log Z$ ESTIMATOR ANALYSIS

The bound in Equation 17 could be derived in the following way:

$$\log Z = \log \int \mu(\mathbf{x}_N) \, \mathrm{d}\mathbf{x}_N = \log \int \mathcal{P}(\mathbf{x}_{0:N}) \, \mathrm{d}\mathbf{x}_{0:N} = \log \mathbb{E}_{\mathcal{Q}} \left[ \frac{\mathcal{P}(\mathbf{x}_{0:N})}{\mathcal{Q}(\mathbf{x}_{0:N})} \right] \tag{39}$$

$$= \log \mathbb{E}_{\mathcal{Q}} \left[ \frac{1}{B} \sum_{b=1}^{B} \frac{\mathcal{P}(\mathbf{x}_{0:N}^{(b)})}{\mathcal{Q}(\mathbf{x}_{0:N}^{(b)})} \right] \leq \mathbb{E}_{\mathcal{Q}} \left[ \log \frac{1}{B} \sum_{b=1}^{B} \frac{\mathcal{P}(\mathbf{x}_{0:N}^{(b)})}{\mathcal{Q}(\mathbf{x}_{0:N}^{(b)})} \right] \tag{40}$$

$$\approx \log \frac{1}{B} \sum_{b=1}^{B} \exp \left( S(\tau^{(b)}) \right), \quad \tau^{(b)} \sim \mathcal{Q}. \tag{41}$$

Notice that this bound is valid without any assumption on $\mathcal{Q}$. According to analysis in Burda et al. (2016), this lower bound estimator is monotonically increase as $B$ goes larger; what's more, it is a tight bound which means that it would converge to $\log Z$ when $B \to \infty$. We remark that this is actually also the estimator used in Zhang & Chen (2022).

### C.3 GRADIENT VARIANCE ANALYSIS

In Figure 2, we conduct experiment on Funnel and report the gradient variance of DGFS and PIS. The variance of DDS is in similar scale to PIS thus we ignore it. According to the analysis in Section 3.3, there are two reasons of why DGFS is a more robust algorithm: (1) DGFS utilize short trajectory which is beneficial for efficient credit assignment (2) DGFS does not take the KL divergence minimizing formulation which has non-zero gradient variance. To disentangle these two factors, we run a variant of DGFS which only utilize full trajectories to update the parameters in Figure 7. This variant is shown to have gradient variance higher than the original DGFS but still much lower than PIS. This confirms that both factors are valid reasons for why DGFS has a lower gradient variance.

For general probabilistic methods with KL between model $q_{\boldsymbol{\theta}}(\cdot)$ and target distribution $p(\cdot)$,

$$\nabla_{\boldsymbol{\theta}} \mathcal{D}_{\mathrm{KL}} \left( q_{\boldsymbol{\theta}}(\mathbf{x}) \| p(\mathbf{x}) \right) = \mathbb{E}_q \left[ \nabla_{\boldsymbol{\theta}} \log q_{\boldsymbol{\theta}}(\mathbf{x}) \log \frac{q_{\boldsymbol{\theta}}(\mathbf{x})}{p(\mathbf{x})} \right] + \mathbb{E}_q \left[ \nabla_{\boldsymbol{\theta}} \log q_{\boldsymbol{\theta}}(\mathbf{x}) \right]. \tag{42}$$

When the model reaches its optimal solution, i.e., $q_{\boldsymbol{\theta}} = p$, the first term is zero as it contains $\log \frac{q_{\boldsymbol{\theta}}(\mathbf{x})}{p(\mathbf{x})} = 0$. However, the second term $\nabla_{\boldsymbol{\theta}} \log q_{\boldsymbol{\theta}}(\mathbf{x})$ *is only zero under expectation of* $q_{\boldsymbol{\theta}}$. That is to say, there will be a stochastic non-zero gradient term with KL formulation. This observation is the core motivation of Roeder et al. (2017).

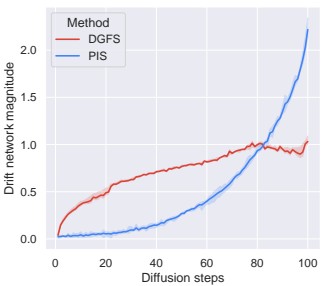

Figure 6: Drift network magnitude of DGFS and PIS.

Figure 7: Gradient variance of DGFS, PIS, and DGFS trained with only full paths.

Figure 8: Manywell results.

---

**Algorithm 1** DGFS training algorithm

---

**Require:** DGFS model with parameters $\boldsymbol{\theta}$, reward function $\mu(\cdot)$, variance coefficient $\tilde{\sigma}$.
  **repeat**
    Sample $\tau$ with control $\mathbf{f}(\cdot, \cdot; \boldsymbol{\theta})$ and $\tilde{\sigma}$;
    $\triangle\boldsymbol{\theta} \leftarrow \nabla_{\boldsymbol{\theta}}\mathcal{L}(\tau; \boldsymbol{\theta})$ (as per Equation 15);
    Update $\boldsymbol{\theta}$ with Adam optimzier;
  **until** some convergence condition

---

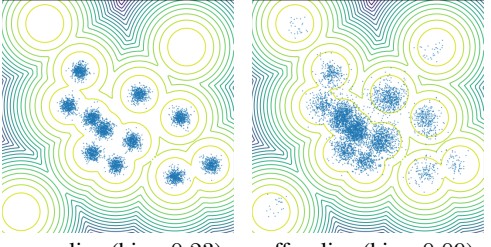

on-policy (bias=0.23)    off-policy (bias=0.09)

Figure 9: Demonstration of DGFS's off-policy exploration ability in a modified MoG+ task.

## C.4 DGFS DETAILS

Here we describe details about DGFS algorithm. As shown in Algorithm 1, DGFS first interact with the environment to collect training data. Next, it updates its parameters $\boldsymbol{\theta}$ with methods described in Section 3. Notice that the training of DGFS is very general and does not have to use on-policy trajectories. Therefore, we could use a different variance coefficient $\tilde{\sigma}$ to rollout. This could potentially bring better exploration ability and thus better mode capturing performance. In this work, we assume DGFS does on-policy rollout unless otherwise specified.

We use PyTorch to implement DGFS algorithm. For PIS implementation, we take its original PyTorch code base. For DDS, we build our own implementation based on PIS code base. For other methods, we take the DeepMind jax code base.

DGFS contains two sets of deep network parameters: one is the same as the neural network in PIS / DDS, the other is the flow function to amortize the intermediate marginal densities. The former one is for the control network, and express it in the way of $\mathbf{f}(\cdot, n)/\sigma = \mathrm{NN}_1(\cdot, n), +\mathrm{NN}_2(n)\cdot\nabla\log\mu(\cdot)$. These use Fourier features to embed the step index input. We also ablate diffusion-based sampling methods without gradient information in network parametrization in C.6. The second one is a scalar output network $\mathrm{NN}(\cdot, n)$ with similar structure. Same to PIS, we use two 64-dimensional hidden layers after the embedding layer. We set $\lambda$ to be 2. For all experiments, we train with a batch size of 256 and Adam optimizer. We have not tuned too hard on the learning rate, but simply use $1 \times 10^{-4}$ and $1 \times 10^{-3}$ for the policy network (*i.e.*, drift network) and the flow function network. The training keeps for 5000 optimization steps although almost all experiments converge in the first 1000 steps. Similar to PIS, we set the number of diffusion steps $N$ to 100 for all experiments. We set the diffusion step size $h$ to 0.05 for the MoG and Cox task and 0.01 for other tasks for all diffusion based methods. We set $\sigma$ to 1. Zhang & Chen (2022) set evaluation particle number $B = 6000$ for the funnel task and $B = 2000$ for other tasks for unknown reasons. In this work, we set the number of evaluation particles $B$ to be 2000 for all tasks for consistency.

## C.5 MORE EXPERIMENTS

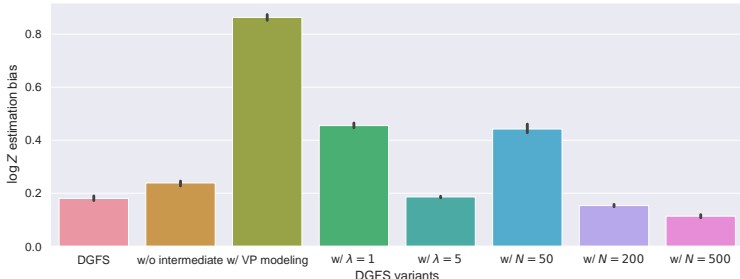

Figure 10: Ablation on DGFS design choices on the VAE task.

Table 2: Experimental results on benchmarking target densities, where the diffusion-based samplers are not allowed to directly access gradient information of the target log density in the drift network design. Note the PIS-NN and DDS-NN here still require to know the gradient information due to their KL formulation, while the DGFS-NN here is a real black-box algorithm and only need to know zeroth order information from the target.

|                          | MoG                | Funnel             | Manywell           | VAE                | Cox                |
|--------------------------|--------------------|--------------------|--------------------|--------------------|--------------------|
| PIS-NN (w/ gradient)     | $1.943_{\pm0.259}$ | $0.374_{\pm0.024}$ | $2.712_{\pm0.017}$ | $0.368_{\pm0.064}$ | $85.36_{\pm3.342}$ |
| DDS-NN (w/ gradient)     | $2.183_{\pm0.018}$ | $0.351_{\pm0.022}$ | $4.056_{\pm1.195}$ | $1.740_{\pm1.158}$ | N/A                |
| DGFS-NN                  | $0.023_{\pm0.001}$ | $0.354_{\pm0.072}$ | $2.659_{\pm0.057}$ | $0.496_{\pm0.196}$ | $109.0_{\pm4.344}$ |

We conduct ablation studies on a series of design choices of DGFS. In Figure 10, we compare the performance of DGFS and a few its variants on the VAE task. In the VAE task, the unnormalized target density $\log \mu(\cdot)$ is defined by the log latent prior $\log p(\cdot)$ plus the log decoder likelihood $\log p(\mathbf{x}_{\text{obs}}|\cdot)$, which equals the latent posterior $\log p(\cdot|\mathbf{x}_{\text{obs}})$ plus a constant value. The VAE is trained on the MNIST dataset, and $\mathbf{x}_{\text{obs}}$ is a fixed image data. We first remove the usage of the intermediate signals (*i.e.*, the parameterization of flow function in Equation 16), which leads to slightly worse performance. We then try to train DGFS with VP modeling in Appendix B.2. For this VP modeling, we also take the form of Equation 16 for getting intermediate signals. This gives a bias of $0.86$. Notice that this is still better than both PIS and DDS, demonstrating the validity of the proposed DGFS training method. We also ablate the weighting coefficient $\lambda$ and number of diffusion steps $N$. We can see their changes lead to small variations but the overall convergence remains good.

Regarding the running time of the proposed algorithm, DGFS has the same inference procedure as PIS, and thus having the same inference speed as PIS. The training overhead of DGFS is slightly ($20\%$) higher than PIS due to the introduction of the additional flow function network. In the Funnel task, the training time of DGFS and PIS for one batch are $0.54$s and $0.65$s respectively.

We then show some missing results from the main text. We put the visualization of the joint of the second and third dimension in Manywell task in Figure 8. We can see that PIS misses one important mode and DDS could not handle the mode separation well enough.

We tried to use a MNIST-pretrained normalizing flow as the target density. However, we find a well-trained normalizing flow almost always assign $-\infty$ to the log likelihood for out-of-distribution samples, which makes it impossible for any sampler to success. This actually makes sense as the normalizing flow is only trained on a finite size image dataset and never gets out-of-domain inputs.

## C.6 Remove gradient information from parametrization

In the network design, PIS explicitly use the score function—gradient of the log unnormalized density function—as one term in the drift network parametrization. Our work and DDS follow its modeling. In this section, we show performance where the score function are removed from the architecture of all three diffusion-base samplers in Table 2. We attach "-NN" postfix to their names to distinguish between their original algorithms. Notice PIS-NN and DDS-NN still require the knowledge of the score function implicitly based on their KL formulation. On the other hand, DGFS-NN

Table 3: Experimental results on benchmarking target densities. DGFS+ denotes DGFS with additional off-policy exploration trick in the initial stage of training.

|       | MoG | Funnel | Manywell | VAE | Cox |
|-------|-----|--------|----------|-----|-----|
| DGFS  | $0.019_{\pm0.008}$ | $0.274_{\pm0.014}$ | $0.904_{\pm0.067}$ | $0.180_{\pm0.083}$ | $8.974_{\pm1.169}$ |
| DGFS+ | $0.012_{\pm0.002}$ | $0.322_{\pm0.029}$ | $0.932_{\pm0.123}$ | $0.162_{\pm0.015}$ | $10.23_{\pm1.237}$ |

Table 4: Experimental results on benchmarking target densities for methods in Lahlou et al. (2023), with or without off-policy exploration technique. We also present performance of TB and its off-policy variant; see details in Section C.8.

|                       | MoG | Funnel | Manywell | VAE | Cox |
|-----------------------|-----|--------|----------|-----|-----|
| Lahlou et al. (2023)  | $1.891_{\pm0.042}$ | $0.398_{\pm0.061}$ | $3.669_{\pm0.653}$ | $4.563_{\pm1.029}$ | $2728._{\pm51.54}$ |
| Lahlou et al. (2023)+ | $0.024_{\pm0.189}$ | $0.373_{\pm0.020}$ | $6.257_{\pm2.449}$ | $2.529_{\pm0.737}$ | $2722._{\pm31.64}$ |
| TB                    | $0.018_{\pm0.002}$ | $0.309_{\pm0.023}$ | $1.218_{\pm0.194}$ | $1.097_{\pm0.239}$ | $138.6_{\pm7.850}$ |
| TB+                   | $0.009_{\pm0.003}$ | $0.373_{\pm0.031}$ | $1.210_{\pm0.192}$ | $0.842_{\pm0.248}$ | $240.4_{\pm0.998}$ |

is a real black-box method, *i.e.*, only requires zeroth order information of the unnormalized density function. Despite using less information, DGFS-NN is still very competitive in most of the tasks.

## C.7 Off-policy exploration ability of DGFS

The usage of the KL divergence in previous works restricts the training to be on-policy, as the training trajectory has to come from the model itself. Using off-policy samples is possible only with importance sampling, which typically results in high variance. This limits the exploration ability of these methods. On the other hand, it is first shown in Malkin et al. (2023) that GFlowNets can potentially benefit from the use of off-policy exploration. To be more specific, as has been shown by a similar analysis in prior works Malkin et al. (2023); Lahlou et al. (2023), the objectives in Equation 9 or Equation 14 do not require the training samples to follow any particular distribution (only to have full support), which means our method supports off-policy training without importance sampling. Therefore, it is possible to employ off-policy exploration by using a larger variance coefficient in the forward process during the rollout stage. In theory, this could help DGFS to better capture diversity in complex multimodal target distributions.

In an algorithmic perspective, we could use a slightly larger variance coefficient $\tilde{\sigma} > \sigma$ in the rollout stage of Algorithm 1 to ensure that information of more modes fall into the training data, which would drive the training of DGFS to put effort on these distant modes. To demonstrate this property, we carefully create a more difficult Mixture of Gaussian task and name it as *MoG+*. We intentionally put a few very distant modes in this MoG+ task, as shown by the contours in Figure 9. We set the exploration variance coefficient $\tilde{\sigma} = 2$ contrary to $\sigma = 1$, which achieves better mode coverage and lower normalizing factor estimation bias with the same neural network capacity.

We conduct an initial experiment to investigate this possibility in DGFS. We test a strategy that linearly anneals the $\tilde{\sigma}$ from 1.1 to 1 during the first 1000 steps of training, and denote it as DGFS+ in Table 3. Our experiment results show that this simple strategy can improve the performance under some scenarios, and can be treated as a potential hyperparameter to extend the application scope of DGFS.

## C.8 Other ablation study

We run the method in Lahlou et al. (2023) in this section. As shown in Table 4, this previous method and its off-policy variant are far from comparable to the baselines in Table 1. We also run the trajectory balance method with gradient information (denoted as TB) and its variant with off-policy exploration (denoted as TB+) and present in the same table. These two methods achieve relatively better performance, but still worse than that of DGFS.

