# OpenReview forum: "Diffusion Generative Flow Samplers: Improving learning signals through partial trajectory optimization"
_ICLR.cc/2024/Conference — ICLR 2024 poster_

### Official Review · Reviewer_hxZf · 2023-10-24

**Soundness:** 4 excellent
**Presentation:** 3 good
**Contribution:** 3 good
**Rating:** 8
**Confidence:** 4

**Summary:**

This paper introduces Diffusion Generative Flow Samplers (DGFS), a neural-based sampling algorithm that utilizes diffusion models. DGFS draws heavy inspiration from GFlowNets (typically constrained to discrete data spaces) and offers a practical and scalable solution for sampling in continuous spaces. The primary message conveyed by this paper is that DGFS demonstrates the ability to leverage partial trajectory segments, thereby enabling a more efficient approach to the learning problem when compared to existing methods, which require learning over complete trajectories (from start to finish), as seen in denoising diffusion samplers (DDS). Consequently, DGFS can sample from target unnormalized probability distributions by updating its parameters during a time-dependent stochastic training process. In other words, it can do so without requiring a full trajectory specification and can accommodate intermediate signals injected between time steps before the sampling process is complete. To accomplish this, a neural network denoted as $F_n(\theta)$ is trained to approximate the unnormalized density of the $n-th$ step.

The authors argue that DGFS should lead to more stable training, reduce variance in gradient computations, and ultimately provide access to informative intermediate training signals.

**Strengths:**

- The paper is well-written, scientifically sound, and highly rigorous, featuring a compelling theoretical framework. The main text is effectively complemented by the appendix materials, which offer additional mathematical details and experimental results.

- I particularly appreciate the table that discusses notation on the first page of the appendix. I believe this should be considered standard practice, as it greatly aids readers in navigating the mathematical content.

- The paper extends the theory of GFlowNets to address sampling problems in continuous space. This extension holds significant relevance for numerous applications, especially in the field of physical and chemical sciences, where unnormalized target densities in continuous space are frequently encountered.

- The concepts of leveraging information from partial trajectory paths and training intermediate layers offer benefits from both theoretical and practical standpoints.

**Weaknesses:**

- **Limited Experiments:** I consider the experiments presented in the paper to be its primary weakness. While the theory is well-presented and comprehensible, the experiments reported in the paper appear to fall short in terms of comparison with more sophisticated baseline methods, particularly in the context of flow-based samplers.

- **Related Works:** Expanding the related work section to include discussions of prior research addressing similar problems, such as gradient variance reduction and efficient sampling of multimodal densities [1, 2, 3], would add significant value. More details are provided below.

**Questions:**

- On the bottom of page 5, the authors state, "On the other hand, notice that our objectives in Equation 9 or Equation 14 do not require the training samples to follow any particular distribution (only have full support)." While I believe that in the asymptotic regime of infinitely many samples this holds, it may not be true for practical cases with a finite number of samples. In such cases, amortized variational inference schemes might encounter issues, as discussed in recent works [4]. I'm interested in whether this assumption can be relaxed or if the authors can provide arguments for the general validity of the 'full support' assumption in their case.

- In the discussion about Variance Gradient Updates (section 3.3A), the authors cite works like Roederer et al. (2017), which reported that the variance of the gradient does not necessarily vanish even when the optimal distribution is achieved. However, in Refs. [1,2], authors use information-theoretical arguments to demonstrate that the variance can indeed approach zero when the target distribution is perfectly learned, using the so-called path gradient method for normalizing flows. It would be interesting to see a comparison of DGFS with such a method (specifically designed for continuous normalizing flows in [2]). Additionally, given the discussion in Sec. 3.3A, it would be beneficial to include these references and expand the discussion.

- The concept of injecting information between a sequence of transformations strongly reminds me of simulated annealing and annealed importance sampling (as seen in Stochastic Normalizing Flow [4,5]). Could the authors expand on this, highlighting the significant similarities and differences, if any? There appears to be a notable overlap worth exploring.

- I'm interested in seeing how the lower bound estimator for the log partition function used in this work compares to the asymptotically unbiased estimator for the partition function proposed in [7,8].

- As pointed out in the **Weaknesses** section, I'm curious about how the results for the many-well task compare to [3], which has reported notable results in data-free training for multimodal target densities for quantum chemical tasks as well as the many-well problem.

**Side remark**: While I acknowledge the significant contribution and potential impact of the paper, I believe additional robust experimental evidence and a broader discussion, as outlined above, would strengthen its acceptance. Adding new baselines, particularly [2,3], to the comparison in Figure 2 and Table 1 would be a valuable addition to the paper.

**Minor:**

- Page 2: below eq (4) the authors state: "It can be shown that the marginal distribution of P at the terminal time N is exactly proportional to the target μ(·).". While it might be self-evident to some, it might be helpful to provide a hint for an explicit derivation in the appendix.

- Page 3: Above eq (7), the authors mention, "[…] stochastic optimal control formulation seen in prior works." It would be beneficial to include explicit references to these prior works.

- Page 7: to the best of my knowledge, what the authors call "Hamilton Monte Carlo" is more often found in the literature as "Hamiltonian Monte Carlo".

- Page 7: In the related work section, the authors might consider citing some of the references listed below, being relevant within the context of NF-based samplers. Notably, some of these references have shown how to derive an asymptotically unbiased estimator for the normalizing constant [7,8,9], also known as the partition function in the realm of Boltzmann distributions. This is a quantity of interest for the present work as well.

**References:**

- [1] [Vaitl, Lorenz, et al. "Gradients should stay on path: better estimators of the reverse-and forward KL divergence for normalizing flows." Machine Learning: Science and Technology 3.4 (2022): 045006](https://iopscience.iop.org/article/10.1088/2632-2153/ac9455/pdf)
- [2] [Vaitl, Lorenz, et al. "Path-gradient estimators for continuous normalizing flows." International Conference on Machine Learning. PMLR, 2022.](https://proceedings.mlr.press/v162/vaitl22a/vaitl22a.pdf)
- [3] [Midgley, Laurence Illing, et al. "Flow annealed importance sampling bootstrap." arXiv preprint arXiv:2208.01893 (2022).](https://arxiv.org/pdf/2208.01893)
- [4] [Wu, Hao, Jonas Köhler, and Frank Noé. "Stochastic normalizing flows." Advances in Neural Information Processing Systems 33 (2020): 5933-5944.](https://proceedings.neurips.cc/paper/2020/hash/41d80bfc327ef980528426fc810a6d7a-Abstract.html)
- [5] [Caselle, Michele, et al. "Stochastic normalizing flows as non-equilibrium transformations." Journal of High Energy Physics 2022.7 (2022): 1-31.](https://arxiv.org/pdf/2201.08862.pdf)
- [6] [Nicoli, Kim A., et al. "Detecting and Mitigating Mode-Collapse for Flow-based Sampling of Lattice Field Theories." arXiv preprint arXiv:2302.14082 (2023).](https://arxiv.org/pdf/2302.14082)
- [7] [Nicoli, Kim A., et al. "Asymptotically unbiased estimation of physical observables with neural samplers." Physical Review E 101.2 (2020): 023304.](https://link.aps.org/accepted/10.1103/PhysRevE.101.023304)
- [8] [Wirnsberger, Peter, et al. "Targeted free energy estimation via learned mappings." The Journal of Chemical Physics 153.14 (2020).](https://pubs.aip.org/aip/jcp/article/153/14/144112/316574)
- [9] [Nicoli, Kim A., et al. "Estimation of thermodynamic observables in lattice field theories with deep generative models." Physical review letters 126.3 (2021): 032001.](https://link.aps.org/pdf/10.1103/PhysRevLett.126.032001)

---

> ### Author Response · Authors · 2023-11-17
>
> Thank you for your insightful review. We have updated the writing of our submission and we sincerely hope the following will answer your points and further improve the score.
>
> **Regarding experiment limitation.** We understand the reviewer's point about the limitations in our experiments. To address this, we have carried out further experiments, which we hope will answer the reviewer's concerns. The details are provided below.
>
>
> **Regarding related works.** We appreciate the reviewer's suggestion to include additional references. We have read these papers and agree that they are very related to our paper, therefore we have added a discussion about prior works [1-3, 5-9] in the related work section in the updated draft. We believe these additions have enriched our writing by providing more descriptions especially on normalizing flow based methods.
>
>
> > On the bottom of page 5, the authors state, "On the other hand, notice that our objectives in Equation 9 or Equation 14 do not require the training samples to follow any particular distribution (only have full support)." While I believe that in the asymptotic regime of infinitely many samples this holds, it may not be true for practical cases with a finite number of samples. In such cases, amortized variational inference schemes might encounter issues, as discussed in recent works [4]. I'm interested in whether this assumption can be relaxed or if the authors can provide arguments for the general validity of the 'full support' assumption in their case.
>
> Your statement is grounded in prior GFlowNet theory [10, 11]. These theories propose that when a GFlowNet achieves zero training loss across the board, it is assured to sample objects with probability proportional to the given reward (density) function. In essence, the learned probabilistic model is capable of generating samples that perfectly align with the target density. For additional details, we direct you to Section 3.2 in [10] and Section 3.4 in [11]. In our paper, we are actually making the assumption that, given a neural network with infinite capacity and training data encompassing the entire space, the network can be trained to achieve zero training loss. As per GFlowNet theory, under the conditions that the GFlowNet model possesses sufficient capacity and the training trajectories are fully supportive, the GFlowNet can be accurately trained to sample. However, in practical scenarios, both two assumptions are not fully met: neural network architectures have limited expressiveness, and training data coverage is incomplete. When these assumptions are relaxed, it is no longer certain that our model will still learn to sample correctly. In real-world applications, our aim is to update the model efficiently with limited training data. A key insight, therefore, is to ensure that the training trajectories cover significant modes of the target distribution, reflecting the most important aspects despite data limitations.
>
>
> > In the discussion about Variance Gradient Updates (section 3.3A), the authors cite works like Roederer et al. (2017), which reported that the variance of the gradient does not necessarily vanish even when the optimal distribution is achieved. However, in Refs. [1,2], authors use information-theoretical arguments to demonstrate that the variance can indeed approach zero when the target distribution is perfectly learned, using the so-called path gradient method for normalizing flows.
> > It would be interesting to see a comparison of DGFS with such a method (specifically designed for continuous normalizing flows in [2]).
> > Additionally, given the discussion in Sec. 3.3A, it would be beneficial to include these references and expand the discussion.
>
> Thank you for bringing up the important work of [1,2], which we have indeed missed in the initial writing. Similar to the analysis in Roeder et al. (2017), [1,2] propose to decompose the total derivative into the summation of a path-gradient term and a score term, and only conisder the path-gradient term when calculating loss objectives. The path-gradient term indeed has zero gradient variance at the optimal solution. We think this is very related to our work, thus we have added reference and discussion in the updated draft.
>
> We agree with the reviewer's suggestion for doing experimental comparison, so we implement the path-gradient version of neural ODE algorithm in [2] with PyTorch. We train the neural ODE to learn to sample from the unnormalized densities in the benchmark in our submission. On the manywell task, this method obtains a log normalizing factor bias of 2.577, which is outperformed by DGFS. We also find it challenging to scale to hard tasks. We are performing more experiments on this algorithm and will add this baseline into the final version of our work. For completeness we provide our anonymous implementation in [this](https://file.io/vEQo2YFry6vv) link.

---

> ### Author Response · Authors · 2023-11-17
> **Official Comment by Authors, 2nd part**
>
> > The concept of injecting information between a sequence of transformations strongly reminds me of simulated annealing and annealed importance sampling (as seen in Stochastic Normalizing Flow [4,5]). Could the authors expand on this, highlighting the significant similarities and differences, if any? There appears to be a notable overlap worth exploring.
>
> We agree that modeling complex distribution through a sequence of transformations is an old yet powerful idea. Representative works include annealed importance sampling [12] and sequential Monte Carlo sampler [13], followed by many learning augmented variants [4,5]. These methods explicitly define a series of densities by linearly interpolating between a reference distribution (usually a standard Gaussian) and the target density in the log probability scale. These methods then design different kinds of transformations between these intermediate level; these transformations are usually complex and are based on MCMC kernels, sometimes involving normalizing flows and rejection-acceptance steps. On the other hand, in our work and other diffusion-based sampling methods, we do not explicitly define the densities in the intermediate level. Instead, we define clear conditional probability for the transition between adajacent levels. In our case, they are all simple Gaussian distributions. We are thus not able to know the exact density functions in the intermediate levels, and thus we use an amortized way to approximate them (Eq. 11-12). In summary, the former kind of modeling explicitly define the intermediate densities and implicitly define the transformations, while the latter modeling does the opposite.
>
>
>
> > I'm interested in seeing how the lower bound estimator for the log partition function used in this work compares to the asymptotically unbiased estimator for the partition function proposed in [7,8].
>
> Thank you for pointing out references [7, 8], which we have now included in our revised draft. The asymptotically unbiased estimator in these works relates to Eq. 17 in the DGFS paper, in that both utilize importance sampling to estimate the normalizing factor. In [7, 8], the proposal is a learned normalizing flow. In our work, the proposal is the forward process
> $\mathcal{Q}$ described in Eq. 1, and our aim is to calculate the integration of the target process $\mathcal{P}$, which has the same value to the integration of the unnormalized density function. We have incorporated a discussion about these papers in Section 3.2 of our updated draft.
>
>
>
> > As pointed out in the Weaknesses section, I'm curious about how the results for the many-well task compare to [3], which has reported notable results in data-free training for multimodal target densities for quantum chemical tasks as well as the many-well problem.
>
> Thank you for bringing up this work. We do have include this work in our original writing, and we point to Appendix C.5 for our results.
>
>
>
>
> **Regarding minor points.** Thank you for pointing out these points. We have fixed them in the revision.
>
>
>
> [1] Gradients should stay on path: better estimators of the reverse-and forward KL divergence for normalizing flows
>
> [2] Path-gradient estimators for continuous normalizing flows
>
> [3] Flow annealed importance sampling bootstrap
>
> [4] Stochastic normalizing flows
>
> [5] Stochastic normalizing flows as non-equilibrium transformations
>
> [6] Detecting and Mitigating Mode-Collapse for Flow-based Sampling of
> Lattice Field Theories
>
> [7] Asymptotically unbiased estimation of physical observables with neural samplers
>
> [8] Targeted free energy estimation via learned mappings
>
> [9] Estimation of thermodynamic observables in lattice field theories with deep generative models
>
> [10] GFlowNet Foundations
>
> [11] A theory of continuous generative flow networks
>
> [12] Annealed importance sampling
>
> [13] Sequential Monte Carlo sampler
>
>
> Thank you for your constructive review. Your suggestions have been instrumental in refining our paper, and we hope that our response can increase your score evaluation of our work. Please let us know if there is any additional thing we can do to improve your evaluation of our work.

---

> ### Author Response · Authors · 2023-11-20
> **Looking forward to your response**
>
> Dear reviewer hxZf,
>
> Thanks again for your detailed comments. We hope our responses have addressed your concerns, and thus request for a reconsideration of the score. We would appreciate it if we can get your further feedback at your earliest convenience.

---

> ### Author Response · Authors · 2023-11-21
> **Request for an update of reviews for our submission**
>
> Dear reviewer hxZf,
>
> We thank for your effort for reviewing our paper. We have demonstrated the benefit of our method and the improvement upon other methods. Since the deadline of discussion period is about to end, and other reviewers have given their feedbacks, we are very expectant to see your re-evaluation of our latest replies.
>
> Thank you very much!

---

> ### Comment · Reviewer_hxZf · 2023-11-22
> **Response to authors**
>
> Dear Authors,
>
> My apologies for the delayed reply. I unfortunately got very sick since last week, and I haven't been able to carefully re-read your manuscript and your replies until now.
>
> I am sorry for having let you wait.
>
> **Feedback on the revised version**
>
>
> I appreciate the great effort the authors have put into re-writing part of the paper to enhance clarity and add further ablation studies and experiments. Most of my concerns have been addressed (I still have to wrap my head around some concepts though I believe this does not depend on the paper itself). For this reason, I am happy to increase my score. In particular, I find the broader discussion of related works very useful, as it would help the general audience to contextualize the present work.
>
> I believe the extensive analysis and efforts in re-implementing the path-gradient estimator from Vaitl et al. will substantially improve the clarity of the paper in comparison to existing methods. I look forward to seeing the extended analysis in the final version of this work.

---

> > ### Author Response · Authors · 2023-11-22
> >
> > Thank you for your timely response. We hope you are getting better now. We appreciate your suggestions and believe they will substantially improve our work. Please feel free to reach out to us if there is any further question.

---

### Official Review · Reviewer_Tb6D · 2023-10-31

**Soundness:** 4 excellent
**Presentation:** 3 good
**Contribution:** 2 fair
**Rating:** 8
**Confidence:** 4

**Summary:**

The paper considers the problem of sampling from a distribution given its unnormalized density. Recent work proposes training a diffusion process sampler to match the target distribution. At present such methods are trained by minimizing the Kullback-Leibler (KL) divergence computed using terminal states of the diffusion process as samples. As a result, complete trajectories need to be sampled during training, while only the terminal states get direct learning signal. In this paper, authors propose learning an additional neural net to provide learning signal for intermediate steps. Authors note that the resulting objective matches the detailed balance objective in GFlowNet literature, and tap into this literature to further improve the learning objective. Authors evaluate the method on several target distributions, showing improved performance compared to common baselines, including existing diffusion-based samplers (PIS/DDS).

**Strengths:**

Significance: diffusion models are state of the art in generative modelling, where we can train them efficiently using the step-wise de-noising objective. Finding similarly effective objectives for training diffusion models as samplers is an important research direction.

Motivation/soundness: the method is well-motivated and theoretically sound. Authors start with a clear problem with existing diffusion-based samplers (weak training signal at intermediate steps), and propose to define intermediate targets for the diffusion process using a neural network trained in parallel. The learning objective for the additional network neatly avoids having to estimate the integral in Eq. (11). The developed connection with GFlowNets allows the method to benefit from developments in this literature.

Results: authors evaluate the propose method on several benchmarks, including the high-dimensional Cox distribution. A good selection of baselines is used, including MCMC, methods based on normalizing flows, and diffusion-based methods. The proposed method demonstrates improved performance across the board, as well as less variance in its results when compared to existing diffusion-based methods.

Quality: the write-up is of high quality, with only a few minor typos, clean mathematical notation, and high-quality figures.

**Weaknesses:**

Originality/novelty: continuous GFlowNets and their connection to diffusion models have been explored by Lahlou et al. (2023) and Zhang et al. (2022a/2023b). The same authors have explored training diffusion models using alternative consistency-based objectives. While the setting in this paper is different (other work considers generative modeling, not sampling), it raises questions about the novelty of the method. Additional discussion with crisp comparisons to Lahlou et al./Zhang et al. would help.

Significance of the GFlowNet connection: Figure 10 in the appendix suggests that the effect of the forward-looking trick is relatively minor. It would be useful to see the effect of using Eqs. (14-15) instead of the originally proposed objective in Eq. (12). In other words, the practical effect of the GFlowNet connection (and associated improvements) is not clear from the results.

Clarity: Sections 2 and 3 are difficult to parse in places. The GFlowNet description in Section 2.2 is extremely dense. The separation of "learning from intermediate steps" vs. "learning with incomplete trajectories" in Section 3 is confusing. More generally, Section 3 could be re-structured to separate proposed methods, existing methods, and discussion more clearly. Section 4 is dense, with a lot of the references already introduced earlier in the paper.

**Questions:**

At the end of page 5 authors say that the method does not "require the training sample to follow any particular distribution (only to have full support)". Have authors measured the effect of choosing the sampling policy? Is it worthwhile to try to explore intelligently?

---

> ### Author Response · Authors · 2023-11-17
>
> Thank you for your insightful review. We have updated the writing of our submission and we hope the following will answer your points.
>
> > Additional discussion with crisp comparisons to Lahlou et al. [2] / Zhang et al. [1] would help.
>
> Thank you for suggesting a discussion on related works. We clarify that [2] is an important precursor to DGFS, as it first established a sound theoretical framework for continuous state space GFlowNets and conducted an initial numerical study on sampling simple densities. DGFS extend on [2] and propose more scalable algorithm that obtain better performance in challenging benchmarks. [1] is another important prior work that initially established the connection between GFlowNets and diffusion models, a foundational concept that [2] also relies upon. [1] conducts experiments in the generative modeling setup with given datasets, training diffusion models using GFlowNet-inspired consistency-based methods, but it does not experiment with sampling given unnormalized densities like [2] and this work.
> We put high value on thorough discussion with prior works, and we have update our introduction section and related work section accordingly to clarify these connections.
>
> [1] Unifying generative models with GFlowNets. ("Zhang et al. (2022)")
>
> [2] A theory of continuous generative flow networks. ("Lahlou et al. (2023)")
>
>
> > It would be useful to see the effect of using Eqs. (14-15) instead of the originally proposed objective in Eq. (12). In other words, the practical effect of the GFlowNet connection (and associated improvements) is not clear from the results.
>
>
> To better show the effect, we implement an algorithm based on Eq. 12 rather than Eq. 15.  We keep the same training architecture, hyperparameters, and evaluation protocol for a fair comparison. This includes the use of the technique in Eq. 16 to incorporate intermediate signals. In the manywell task, this algorithm achieves a log normalizing factor estimation bias of 1.082 ± 0.061 compared to 0.904±0.067 of DGFS. We remark that this is also an competitive result compared to other baseline methods. We will add this result to the final version of this paper.

---

> > ### Comment · Reviewer_Tb6D · 2023-11-20
> >
> > I thank the authors for their response. I have found the comparison to prior work to not be very precise still. Some concrete clarifying questions below.
> >
> > > DGFS extend on [2] and propose more scalable algorithm that obtain better performance in challenging benchmarks.
> >
> > Could you clarify what "more scalable" means in this context? Why is DGFS more scalable? What evidence do authors have for the "better performance" claim: have they compared DGFS to the methods of Lahlou et al.?
> >
> > > [1] conducts experiments in the generative modeling setup with given datasets, training diffusion models using GFlowNet-inspired consistency-based methods, but it does not experiment with sampling given unnormalized densities like [2] and this work.
> >
> > How do these "GFlowNet-inspired consistency-based methods" differ from DGFS? Are these methods not applicable to sampling, or have Zhang et al. simply not evaluated them on sampling?

---

> > > ### Author Response · Authors · 2023-11-20
> > >
> > > We thank the reviewer for the response and would like to clarify potential misunderstanding.
> > >
> > > > What evidence do authors have for the "better performance" claim
> > >
> > > Yes, we have conduct detailed comparisons in our paper to support this claim. See the following table about performance of [2] and our method. We put more details in Appendix C.8 of our paper.
> > >
> > > |       | MoG         | Funnel      | Manywell    | VAE         | Cox         |
> > > |-------|-------------|-------------|-------------|-------------|-------------|
> > > |  [2]    | 1.891 ± 0.042    | 0.398±0.061     | 3.669 ± 0.653    | 4.563 ± 1.029     | 2728. ± 51.54      |
> > > |  [2]  w/ off policy | 0.024 ± 0.189    |  0.373 ±0.020 | 6.257 ± 2.449    | 2.529 ± 0.737     | 2722. ± 31.64      |
> > > | DGFS  | 0.019±0.008 | 0.274±0.014 | 0.904±0.067 | 0.180±0.083 | 8.974±1.169 |
> > >
> > > > How do these "GFlowNet-inspired consistency-based methods" differ from DGFS?
> > >
> > > These methods differ significantly from DGFS, both in the setups and the algorithmic specifics. As the reviewer has pointed out, they are not applicable to sampling and thus not able to be evaluated on sampling tasks. In fact, they are appropriate for the setting where the dataset is given, which is usually the case in the generative modeling setup. In terms of algorithmic details, the consistency-based method introduces a regularization technique that does not require the access to the density function to learn the backward policy, while DGFS is designed for learning the forward policy with the density function rather than the given dataset. In DGFS the backward policy is fixed.
> > >
> > > Thank you for your constructive review again! Please let us know if there is any additional thing we can do to enhance your assessment of our work.

---

> > > > ### Comment · Reviewer_Tb6D · 2023-11-21
> > > >
> > > > I thank the authors for further clarifications and additional results. Several of my points have been addressed, hence I increase my score. I encourage the authors to include the additional comparisons, extended discussion of prior work and ablation studies in the final version of the paper, to make the contribution and its context crystal clear.

---

> > > > > ### Author Response · Authors · 2023-11-21
> > > > >
> > > > > Thank you for your prompt response. We have included these discussions and results in the updated version of the paper. If you also have any further question by any chance, please feel free to reach out again.

---

> ### Author Response · Authors · 2023-11-17
> **Official Comment by Authors, 2nd part**
>
> > Clarity: Sections 2 and 3 are difficult to parse in places. The GFlowNet description in Section 2.2 is extremely dense. The separation of "learning from intermediate steps" vs. "learning with incomplete trajectories" in Section 3 is confusing. More generally, Section 3 could be re-structured to separate proposed methods, existing methods, and discussion more clearly. Section 4 is dense, with a lot of the references already introduced earlier in the paper.
>
> Thank you for your suggestions regarding the structure of our writing. We recognize that our introduction of GFlowNet in Section 2.2 is quite dense, a consequence of space constraints; we plan to include a more comprehensive version in the final version of this paper. Regarding the terms 'intermediate steps' and 'learning with incomplete trajectories', we understand the need for greater clarity. 'Intermediate steps' refers to the application of Eq. 16, which sets our method apart from other diffusion-based sampling methods that only derive signals from the terminal step reward. 'Learning with incomplete trajectories' denotes DGFS's ability to update parameters without needing the full specification of complete trajectories. In the final manuscript, we will refine our explanations to more clearly delineate these concepts. Additionally, we intend to expand the related work section to offer a deeper discussion on how our work relates to and differs from previous studies.
>
>
> > At the end of page 5 authors say that the method does not "require the training sample to follow any particular distribution (only to have full support)". Have authors measured the effect of choosing the sampling policy? Is it worthwhile to try to explore intelligently?
>
>
> Thank you for bringing up this point. Here we perform an intial trial.
> During the initial writing of this work, we plan to systematically study the effect of the off-policy ability. However during the deadline rush, as stated in Appendix C.7, we do not have enough time for this and thus we only keep a case study in Appendix C.7. During the rebuttal period, we perform an initial study on this. We linearly decay the $\sigma$ coefficient in DGFS from 1.1 to 1 during the first 1000 training steps, and denote this strategy as DGFS+. We present the results here.
>
> |       | MoG         | Funnel      | Manywell    | VAE         | Cox         |
> |-------|-------------|-------------|-------------|-------------|-------------|
> | DGFS  | 0.019±0.008 | 0.274±0.014 | 0.904±0.067 | 0.180±0.083 | 8.974±1.169 |
> | DGFS+ | 0.012±0.002 | 0.322 ± 0.229 | 0.932±0.123 | 0.162±0.015 | 10.23±1.237 |
>
> As can be seen from this table, this exploration strategy can bring improvement under some scenarios. This indicates that off-policy can be introduced as a hyperparameter to help the sampling. We see this as a promising direction to further improve the performance of our methods in future. We have updated our writing to incorporate this study with details in the Appendix.
>
>
>
> We are thankful for your expert review and would welcome any more advice you could provide to help us improve our work further.

---

### Official Review · Reviewer_oYBq · 2023-10-31

**Soundness:** 3 good
**Presentation:** 3 good
**Contribution:** 4 excellent
**Rating:** 8
**Confidence:** 2

**Summary:**

In this paper, a novel training strategy for diffusion-based samplers is proposed, where the sampler is trained with learning signals for incomplete trajectories. Such a signal is amortized with a novel flow function. Training the sampler with these intermediate learning signals results in reduced variance with respect to similar modes trained only on full trajectories, and shows improved results on a wide set of benchmarks.

**Strengths:**

The paper combines recent improvements to the credit assignment problem for partial trajectories in GFlowNets with recent diffusion-based sampler methods. The introduction of the amortized flow function is novel and well-supported with both theoretical and empirical results. The experimental section presents strong empirical results, as well as insightful analysis of the gradient variance and learned drift function which confirm the claims made by the authors. Overall, the proposed method presents strong performance on challenging tasks, making it a potentially high-impact contribution to the scientific community.

**Weaknesses:**

In my opinion, the paper could improve in terms of clarity and in separating the previous methods from the proposed one. The paper is heavily based on previous contributions, such as denoising diffusion sampler (DDS) and path integral sampler (PIS), as well as recent contributions in GFlowNets like [1,2]. While the authors provide brief explanations of these previous methods, I think it would be beneficial to first give a clearer introduction of what such methods do (especially for DDS and PIS), and then clearly outline how the ideas are combined in DGFS.

The off-policy training strategy is mentioned but could also benefit from additional explanations and perhaps an ablation study, for example, the empirical benefits of training DGFS off-policy vs on-policy. From the text, it is not immediately clear whether DGFS can only be trained off-policy, or if it has the possibility to be trained off-policy as opposed to on-policy.

[1] Madan, Kanika, et al. "Learning GFlowNets from partial episodes for improved convergence and stability." International Conference on Machine Learning. PMLR, 2023.

[2] Pan, Ling, et al. "Better training of gflownets with local credit and incomplete trajectories." arXiv preprint arXiv:2302.01687 (2023).

**Questions:**

In the experiments, there is no comparison with GFlowNets methods. Is that because GFlowNets perform poorly on continuous sampling benchmarks? And on the other hand, can DGFS be used on discrete data space? And if so, how does it perform?

---

> ### Author Response · Authors · 2023-11-17
>
> Thank you for your insightful review. We have updated the writing of our submission and we hope the following will answer your points.
>
>
> > While the authors provide brief explanations of these previous methods, I think it would be beneficial to first give a clearer introduction of what such methods do (especially for DDS and PIS), and then clearly outline how the ideas are combined in DGFS.
>
> Thank you for the suggestion. We agree that our method is indeed based on previous works, which is why we provide an introduction about PIS \& DDS in Section 2.1 and an introduction about GFlowNets in Section 2.2. We acknowledge that this might be limited and not clear enough for the rationale, and we will improve related writing by adding more introductory sections in both the main text and the Appendix in the final version.
>
>
> > The off-policy training strategy is mentioned but could also benefit from additional explanations and perhaps an ablation study, for example, the empirical benefits of training DGFS off-policy vs on-policy. From the text, it is not immediately clear whether DGFS can only be trained off-policy, or if it has the possibility to be trained off-policy as opposed to on-policy.
>
> Thank you for bringing up this point. During the initial writing of this work, we plan to systematically study the effect of the off-policy ability. However during the deadline rush, as stated in Appendix C.7, we do not have enough time for this and thus we only keep a case study in Appendix C.7. We have updated the writing to remove related paragraphs from the main text to ensure the text not being misleading.
>
> During the rebuttal period, we perform an initial study on this. We linearly decay the $\sigma$ coefficient in DGFS from 1.1 to 1 during the first 1000 training steps, and denote this strategy as DGFS+. We present the results here.
>
> |       | MoG         | Funnel      | Manywell    | VAE         | Cox         |
> |-------|-------------|-------------|-------------|-------------|-------------|
> | DGFS  | 0.019±0.008 | 0.274±0.014 | 0.904±0.067 | 0.180±0.083 | 8.974±1.169 |
> | DGFS+ | 0.012±0.002 | 0.322 ± 0.229 | 0.932±0.123 | 0.162±0.015 | 10.23±1.237 |
>
> As can be seen from this table, this exploration strategy can bring improvement under some scenarios. This indicates that off-policy can be introduced as a hyperparameter to help the sampling. We see this as a promising direction to further improve the performance of our methods in future. We have updated our writing to incorporate this study with details in the Appendix.
>
>
>
> > In the experiments, there is no comparison with GFlowNets methods. Is that because GFlowNets perform poorly on continuous sampling benchmarks? And on the other hand, can DGFS be used on discrete data space? And if so, how does it perform?
>
> Previous work [1] has done initial experiment with trajectory balance algorithm on relatively simple continuous sampling tasks to support the validity of continuous GFlowNet theory. The performance in high dimensional tasks of the algorithm is not competitive against the baselines used in this work, as the reviewer supposes. During rebuttal period, we conduct experiment to systematically test the algorithm in [1] and show the results here.
>
> |       | MoG         | Funnel      | Manywell    | VAE         | Cox         |
> |-------|-------------|-------------|-------------|-------------|-------------|
> |  [1]     | 1.891 ± 0.042    | 0.398±0.061     | 3.669 ± 0.653    | 4.563 ± 1.029     | 2728. ± 51.54      |
> |  [1] w/ off policy | 0.024 ± 0.189    |  0.373 ±0.020 | 6.257 ± 2.449    | 2.529 ± 0.737     | 2722. ± 31.64      |
>
>
> All the evaluation protocols and hyperparameter setups are keep consistent with the setup in this work. We can see that the performance of both the variants are far from comparable to the baselines in this work.
>
>
>
> As for discrete data space, we refer to a related previous work [2], where the author use a similar modeling to this work to train a GFlowNet for sampling in a categorical discrete data space. [2] designs a Markovian decision process to sequentially generate categorical data, and use a deep neural network to model the policy that navigates in this decision process. The model can be trained with signals from the reward function (which is parametrized as an energy function in that paper) and is a valid inference machine that can sample with probability proportional to this reward function if the training is complete.
>
>
> [1] A theory of continuous generative flow networks
>
> [2] Generative Flow Neteworks for Discrete Probabilistic Modeling
>
>
> We thank you for your review and we are grateful for any additional insights you might have for our paper.

---

> > ### Comment · Reviewer_oYBq · 2023-11-21
> >
> > I thank the authors for answering my questions and including clarifications to the manuscripts. I confirm my previous score.

---

> > > ### Author Response · Authors · 2023-11-23
> > >
> > > Thank you for your response and the increase of the confidence. If you have any further question, please feel free to raise it.

---

### Official Review · Reviewer_1u4d · 2023-11-01

**Soundness:** 3 good
**Presentation:** 4 excellent
**Contribution:** 3 good
**Rating:** 6
**Confidence:** 3

**Summary:**

The paper proposes an algorithm for better sampling from intractable high-dimensional density functions. The sampling procedure is formulated as a time sequence. The improvement then stems from the usage of partial trajectories instead of merely relying on the end-time variable. Empirical results show the superior performance of the proposed method.

**Strengths:**

I haven't done any research in the sampling area, nor do I have a strong background of the sampling methods and their advantages and disadvantages. However, from reading the paper, I have gained an understanding of the problem to be solved in the paper. Moreover, the methodology makes sense to me. That being said, the paper is well written with a clear logic flow, even for people who is new to the field.

The experiments cover a wide range of data, including high-dimensional settings. The reported results look very promising and indicating the strength of the proposed method.

**Weaknesses:**

I am not very confident in evaluating the novelty of the proposed method. From the introduction, it seems like a combination of the existing ideas in constructing a forward Markov chain and exploitation of the detailed balance in GFlowNets. I would appreciate a discussion regarding the novelty.

What is the running time comparison to existing baseline methods?

The reported results in table 1 uses a different metric for baseline methods. I am curious how does the proposed method compares to baseline methods using the results at the best checkpoint?

**Questions:**

See weakness above.

---

> ### Author Response · Authors · 2023-11-17
>
> Thank you for your insightful review. We have updated the writing of our submission and we hope the following will answer your points.
>
> > From the introduction, it seems like a combination of the existing ideas in constructing a forward Markov chain and exploitation of the detailed balance in GFlowNets. I would appreciate a discussion regarding the novelty.
>
> We appreciate the reviewer's interest in the novelty of our approach. Our work is indeed based on prior GFlowNet foundations  and previous diffusion-based sampling methods. Though the concept of detailed balance is not new, our work is the first to achieve learning without full trajectory specification in the field of continuous sampling from given unnormalized densities with diffusion. This could potentially inspire future sampling research with similar ideas. Furthermore, we propose the use of an intermediate signal (Eq. 16) which is novel and specifically designed for the chosen SDE-based modeling. This largely enhances learning efficiency compared to other diffusion-based sampling methods, which rely solely on terminal signals. Last but not least, our proposed algorithm achieves state-of-the-art performance on challenging sampling benchmarks, demonstrating the significance of our method. In summary, while DGFS integrates existing concepts, its unique contributions lie in its novel training approach, its flexibility in accommodating different formulations, and its solid theoretical backing, all of which mark a significant advancement in the domain of diffusion-based sampling methods.
>
> > What is the running time comparison to existing baseline methods?
>
> We conduct an evaluation to compare the running times of all diffusion modeling methods. On the manywell task, the training time per batch (with a batch size of 256) is 3.24 seconds for PIS, 3.34 seconds for DDS, and 3.74 seconds for DGFS; the inference time for generating a batch (with a batch size of 2000) is 1.52 seconds for PIS, 1.48 seconds for DDS, and 1.50 seconds for DGFS. All tests were performed on a single NVIDIA RTX 8000 GPU. This comparison provides clear insights into the comparable speed of our proposed methods relative to the existing baselines while having better performance.
>
>
> > The reported results in table 1 uses a different metric for baseline methods. I am curious how does the proposed method compares to baseline methods using the results at the best checkpoint?
>
> We conduct experiments on the manywell task. We test every a hundred training step and report the best value for each algorithm. The log normalizing factor estimation bias under this protocol is 22.36 for SMC, 2.079 for VI-NF, 0.055 for CRAFT, 0.088 for PIS, 0.129 for DDS, and 0.058 for DGFS. Notice that there SMC result stays the same as the table, as there is no training involved in it. We remark that the best checkpoint during the training is an unstable metric, as the algorithm's estimate can ocsillate over the course of learning due to randomness. If "good luck" occurs, then a poorly performing algorithm may still obtain an estimate close to the ground truth log normalizing factor at some point during training. Therefore, we use a more stable way in the paper as stated in Section 5.1.
>
>
> We sincerely thank you for your feedback and remain open to any additional suggestions that could further improve our work.

---

> > ### Comment · Reviewer_1u4d · 2023-11-23
> >
> > Thank you for the response and I am willing to keep the positive review and raise my confidence.

---

### Author Response · Authors · 2023-11-17
**General Response**

We extend our sincere thanks to all the reviewers for their insightful feedback. In response, we have revised our draft, highlighting all changes in blue font for ease of reference. The major updates include:


- The addition of detailed discussions on related works. Particularly, in Section 4 we add a discussion of works mentioned by Reviewer **hxZf**. We have also highlight the importance of Lahlou et al. for establishing theoretical framework and initial numerical experiments. These additions significantly enhance the quality of our paper.
- The inclusion of more baseline comparisons and an expanded ablation study, as suggested by the reviewers. This contains a more systematic examination of the off-policy exploration capability.
- Improvements in the clarity and coherence of our writing across multiple sections.

We are happy to have any further discussions that could potentially raise your evaluation of this work.

---

### Public Comment · ~Tiago_Silva4 · 2023-12-02
**Some questions to the authors**

Dear authors,

I am a bit curious about the effective contributions of this work. The use of GFlowNets in stochastic control problems and in diffusion-based modeling, for instance, was already considered by Lahlou et al.'s [1], and the incorporation of intermediate reward signals along a trajectory to improve the training convergence of GFlowNets was thoroughly contemplated in [2]. Moreover, the experimental campaign of the submitted manuscript is very similar to that of Lahlou's work (compare, e.g., Table C.1 of [1] with Table 1 and Figure C.1 of [1] with Figure 5).

Thus, I would like to better understand the methodological contribution of this work. In particular, I believe that the authors’ claimed contribution, namely, the design of a model that learns from incomplete trajectories, lacks a solid theoretical foundation, making it particularly challenging to address the proposed method’s limitations. In this context, can the authors kindly provide a clarification for the following two points?

1. Firstly, the meaning of an intermediate signal $\tilde{R}_{n}$ is mostly unclear; is a constant function an appropriate choice for it? Intuitively, this doesn’t seem to be the case, although, based solely on the paper’s discussion, one cannot answer such a simple inquiry.

2. Secondly, the authors _do_ sample complete trajectories during training (cf. Algorithm 1); I wonder whether the model can learn by exclusively sampling trajectories with limited size. Otherwise, one can hardly say that the model learns from incomplete trajectories.

Furthermore, training a DGFS seems to require computing the target unnormalized density at each transition of the iterative generative process (by Eq. (16)). This implies, for the Cox process, frequently evaluating the density of a 1600-dimensional distribution, which may significantly increase the run-time of DGFS relative to Lahlou et al.’s method [1] (which evaluates the target density only at the end of each trajectory; see Definition 7 of [1]). Thus, it would be interesting to know whether both methods perform similarly (according to the metrics at Tables 3 and 4) given the same time budget.

In summary, I am not sure how this work helps to understand both diffusion models and GFlowNets. The main reason for this is that the work relies on a notion of "intermediate signals", which is very vague (and unoriginal [2]). Notably, the conclusion delegates pinning down this notion to future works ('how can we better design the intermediate local signals in ways more sophisticated than the current straightforward approach?').

[1] A theory of continuous generative flow networks, Salem Lahlou et al., ICML 2023

[2] Better training of GFlowNets with local credit and incomplete trajectories, Ling Pan et al., ICML 2023

---

### Meta-Review · Area_Chair_cPf7 · 2023-12-05

**Metareview:**

The authors consider the problem of sampling from high-dimensional density functions using generative flows to generate approximate samples. They identify that a main obstacle is the training objective requires computing terminal states of the diffusion process, as opposed to the training of diffusion models for generative modeling, which can be considered stagewise, and propose a way to speed up training using partial trajectory segments.

Reviewers found the writing to be clear, the experiments to be convincing and supported by theory.

**Justification For Why Not Higher Score:**

Amount of novelty compared to previous works is not exceptional, as mentioned by some reviewers.

**Justification For Why Not Lower Score:**

All reviewers supported the paper.

---

### Decision · Program_Chairs · 2024-01-16

Accept (poster)